# ROBUSTNESS OF AI-IMAGE DETECTORS: FUNDAMENTAL LIMITS AND PRACTICAL ATTACKS

**Mehrdad Saberi[1], Vinu Sankar Sadasivan[1], Keivan Rezaei[1], Aounon Kumar[1],
Atoosa Chegini[1], Wenxiao Wang[1], Soheil Feizi[1]**

[1]Department of Computer Science, University of Maryland

{msaberi,vinu,krezaei,aounon,atoocheg,wwx,sfeizi}@umd.edu

## ABSTRACT

In light of recent advancements in generative AI models, it has become essential to distinguish genuine content from AI-generated one to prevent the malicious usage of fake materials as authentic ones and vice versa. Various techniques have been introduced for identifying AI-generated images, with watermarking emerging as a promising approach. In this paper, we analyze the robustness of various AI-image detectors including watermarking and classifier-based deepfake detectors. For watermarking methods that introduce subtle image perturbations (i.e., low perturbation budget methods), we reveal a fundamental trade-off between the evasion error rate (i.e., the fraction of watermarked images detected as non-watermarked ones) and the spoofing error rate (i.e., the fraction of non-watermarked images detected as watermarked ones) upon an application of diffusion purification attack. To validate our theoretical findings, we also provide empirical evidence demonstrating that diffusion purification effectively removes low perturbation budget watermarks by applying minimal changes to images. For high perturbation watermarking methods where notable changes are applied to images, the diffusion purification attack is not effective. In this case, we develop a model substitution adversarial attack that can successfully remove watermarks. Moreover, we show that watermarking methods are vulnerable to spoofing attacks where the attacker aims to have real images (potentially obscene) identified as watermarked ones, damaging the reputation of the developers. In particular, by just having black-box access to the watermarking method, we show that one can generate a watermarked noise image, which can be added to the real images, leading to their incorrect classification as watermarked. Finally, we extend our theory to characterize a fundamental trade-off between the robustness and reliability of classifier-based deep fake detectors and demonstrate it through experiments. Code is available at https://github.com/mehrdadsaberi/watermark_robustness.

## 1 INTRODUCTION

As generative AI systems advance in sophistication and accessibility, the production of persuasive fabricated digital content becomes more accessible. These systems have the ability to craft hyper-realistic media forms such as images, videos, and audio (referred to as deepfakes), capable of deceiving viewers and listeners (Helmus, 2022). This misapplication of AI introduces potential hazards related to misinformation, fraud, and even national security issues like election manipulation (Blauth et al., 2022; Chesney & Citron, 2019). Moreover, deepfakes can result in personal harm, spanning from character defamation to emotional distress, impacting both individuals and broader society (Ice, 2019). Consequently, the identification of AI-generated content and, importantly, tracing its sources, emerges as a crucial challenge to address.

Over the years, numerous techniques for recognizing AI-generated images have emerged. Among these, Image watermarking stands out as a promising approach (Honsinger, 2002; Swanson et al., 1998). Watermarking techniques, along with their many other applications (Potdar et al., 2005; Zhao et al., 2023c; Cui et al., 2023), can be integrated with image generation models (Rombach et al., 2022) to inject watermarks to AI-generated images, which enables them to be differentiated from

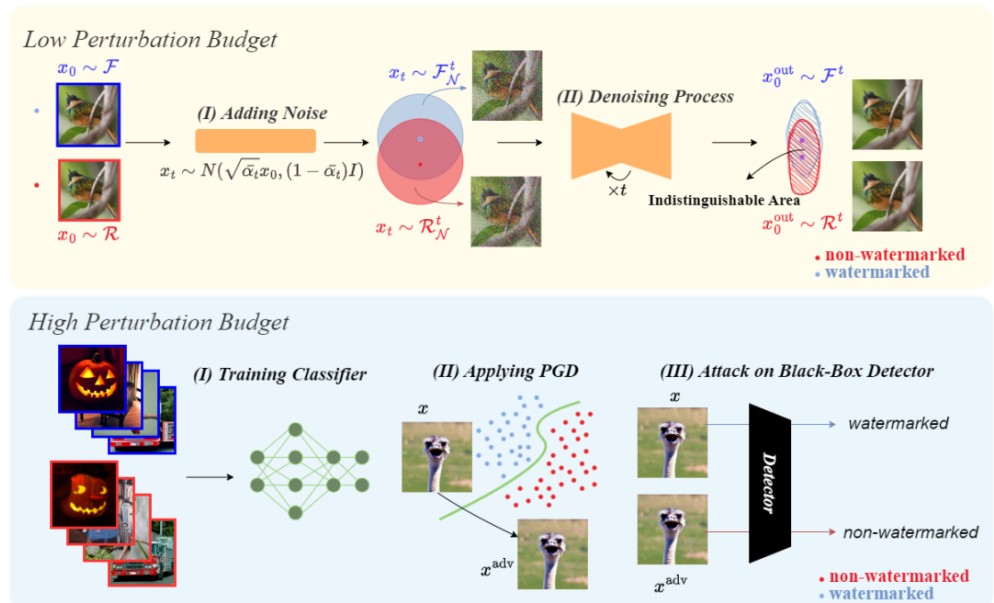

Figure 1: Illustration of our attacks against image watermarking methods. **Upper panel** demonstrates the diffusion purification attack for low perturbation budget (imperceptible) watermarks. It adds Gaussian noise to images, creating an indistinguishable region, which results in a *certified* lower bound on the error of watermark detectors. Noisy images are then denoised using diffusion models. See Section 3.1 for the definition of the used terms (e.g., $\mathcal{R}$, $\mathcal{F}$). **Lower panel** depicts our model substitute adversarial attack against high-perturbation budget watermarks. Our attack involves training a substitute classifier, conducting a PGD attack on the substitute model, and using these manipulated images to deceive the black-box watermark detector.

real images later. These techniques also allow for tracing the source of generation for images. Given the continuous enhancement in deepfake image quality and the growing challenge of distinguishing them from real ones, the adoption of image watermarking over classifier-based detection techniques is becoming a more sensible choice.

In this paper, we demonstrate a fundamental constraint on the robustness of image watermarking methods. We leverage a technique called diffusion purification (Nie et al., 2022), originally proposed as a defense against adversarial examples. This approach involves the introduction of Gaussian noise to images and utilizing the denoising process of diffusion models (Ho et al., 2020) to eliminate the added noise. We offer both theoretical and empirical evidence that this attack amplifies the error rates of watermarking methods that have a low Wasserstein distance between the distributions of their watermarked and non-watermarked images, which we refer to as "low perturbation budget" watermarking methods; i.e., watermarks with subtle image perturbations.

To elaborate, if $\mathcal{R}$ and $\mathcal{F}$ represent the distributions of non-watermarked and watermarked images, and $\mathcal{R}^t$ and $\mathcal{F}^t$ denote the distributions of these images after the application of the diffusion purification attack, we demonstrate that:

$$e_0(\mathcal{F}^t, D) + e_1(\mathcal{R}^t, D) \geq 1 - \mathsf{erf}\left(\frac{\sqrt{\bar{\alpha}_t}\, \mathsf{W}(\mathcal{R}, \mathcal{F})}{2\sqrt{2(1-\bar{\alpha}_t)}}\right),$$

where $e_0$ and $e_1$ correspond to the evasion (type I) and spoofing (type II) errors of detector $D$ (i.e., formally defined in Definition 1), $\mathsf{W}(.,.)$ stands for the Wasserstein distance function, $\mathsf{erf}(.)$ is the Gauss error function, and $\bar{\alpha}_t$ represents the cumulative alpha of the diffusion model at step $t$. To complete our theoretical findings, we empirically show that diffusion purification attack can reduce the AUROC (Area Under the Receiver Operating Characteristic) of some existing low-perturbation watermarks (Zhang et al., 2019c; Cox et al., 2007; Zhao et al., 2023b) to values less than $0.65$ by applying minimal changes to images.

If the Wasserstein distance between the distributions of watermarked and non-watermarked images is large (i.e., high perturbation budget watermarking), our theoretical bound based on diffusion purification attack becomes vacuous. In fact, we also empirically observe that this attack does not

compromise existing high perturbation budget watermarking methods where notable changes are applied to the images such as TreeRing (Wen et al., 2023) (Figure 4). In this regime, we develop a method that trains a substitute classifier capable of distinguishing between watermarked and non-watermarked images. Subsequently, we execute an adversarial attack (Madry et al., 2017) on images using this substitute classifier. Intriguingly, these attacks appear to transfer successfully to the authentic watermark detector. Our adversarial attack manages to decrease the AUROC of the TreeRing method to $0.14$ by employing an $\ell_\infty$ attack with $\epsilon = 2/255$. A distinguishing feature of our attack, in contrast to previously proposed white-box and black-box attacks (Jiang et al., 2023), is that it does not necessitate real-time access to the watermark detector. Instead, it operates by collecting images watermarked by a specific watermark model from the internet.

We note that some watermarking methods such as StegaStamp (Tancik et al., 2020) impose large perturbations in the latent (feature) space but relatively smaller perturbations in the image space (Table 1). We show that both the diffusion purification attack (in the image space) as well as our model substitution adversarial attack are successful in breaking the StegaStamp watermark, especially using larger diffusion steps or adversarial perturbation budgets.

In addition to the previously mentioned attacks, we introduce *a spoofing attack* designed to target the spoofing error in watermarking methods. These attacks have the potential to erroneously categorize explicit or inappropriate content as watermarked, which could have adverse implications for the developers associated with a watermarked generative model, including loss of trust, financial loss, and negative publicity. Our attack functions by instructing watermarking models to watermark a white noise image and then blending this noisy watermarked image with non-watermarked ones to deceive the detector into flagging them as watermarked.

Finally, we extend our theory originally established for watermarking methods, to offer a corresponding theoretical insight for classifier-based AI-image detectors. Our analysis demonstrates a fundamental trade-off between the robustness and reliability of these detectors. As the distributions of real and fake images grow more alike, this trade-off becomes more pronounced. This implies that a detector could only achieve good performance or high robustness, but not both, simultaneously. We further present empirical evidence for this trade-off on some real-world detectors.

**Summary of Contributions.** In this paper, we make the following contributions:

1. We characterize a fundamental trade-off between evasion and spoofing error rates of image watermarking upon the application of a diffusion purification attack. Empirically, we show that diffusion purification attack can break a whole range of watermarking methods that introduce subtle image perturbations (i.e., low perturbation budget image watermarking).

2. For high perturbation image watermarking that leaves notable changes on the original images, we show that the diffusion purification attack is not effective. Instead, we develop a model substitution adversarial attack that can successfully remove the watermarks.

3. We introduce a spoofing attack against watermarking by adding a watermarked noise image to clean images, in order to deceive the detector into flagging them as watermarked.

4. We develop a fundamental trade-off between the robustness and reliability of deepfake detectors and substantiate this concept through experiments.

## 2  PRIOR WORK

**Image Watermarking.** Image watermarking is a versatile technology with applications in copyright protection, content authenticity, data authentication, privacy preservation, and branding. Its evolution began with manual methods like LSB (Wolfgang & Delp, 1996), and later techniques involved altering spatial or frequency domains (Ghazanfari et al., 2011; Holub & Fridrich, 2012; Pevnỳ et al., 2010; Boland et al., 1995; Cox et al., 1996; O'Ruanaidh & Pun, 1997). Various transformations such as DCT, DWT, SVD-decomposition (Chang et al., 2005), and Radon transformations (Seo et al., 2004) were explored. Recent advancements incorporate deep learning and generative models like SteganoGAN (Zhang et al., 2019a), StegaStamp (Tancik et al., 2020) RivaGAN (Zhang et al., 2019c), WatermarkDM (Zhao et al., 2023b), MBRS (Jia et al., 2021), and Tree Ring (Wen et al., 2023), each employing different methods to embed watermarks into images.

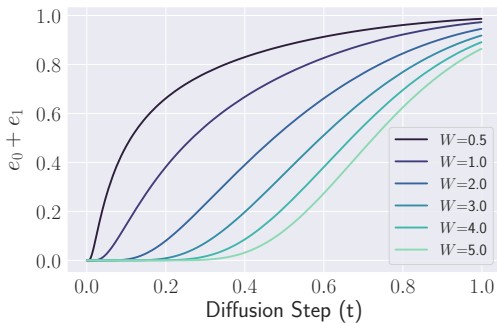 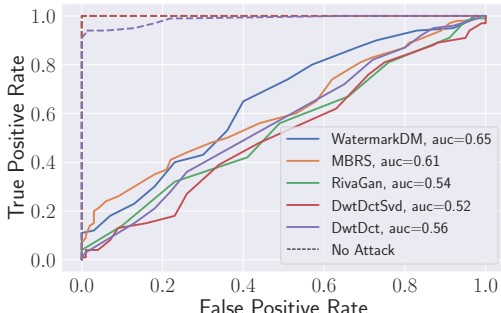

Figure 2: Lower bound on the sum of evasion and spoofing errors of image watermarks against diffusion purification attack from Theorem 1. The beta schedule for the diffusion model is linear in the range $[0.0008, 0.0120]$.

Figure 3: ROC curves for empirical robustness of image watermark methods against diffusion purification attack with $t = 0.2$. The dashed lines show the ROC curves of methods without attacking them.

There have been several works trying to attack watermarking methods (Jiang et al., 2023; Wang et al., 2022a). Notably a recent concurrent work (Zhao et al., 2023a) also proves that the diffusion purification attack is successful against invisible (low perturbation budget) watermarking. However, (Zhao et al., 2023a) is unable to attack high perturbation budget watermarking methods such as Tree Ring or StegaStamp and argues that they are more reliable watermarking alternatives. In contrast, we show that our model substitution adversarial attack can effectively break those watermarking methods. Additionally, we show that several watermarking approaches are vulnerable to spoofing attacks and characterize a robustness-reliability trade-off for a classification-based deepfake detector.

**Classifier-based Detectors.** Several machine-learning approaches focusing on detecting artifacts in AI-generated content have been studied. For instance, Matern et al. (2019) target irregularities in face editing algorithms, while Ciftci & Demir (2019) exploit biological signals. Li et al. (2020) introduce a technique for identifying partially manipulated videos, and Guarnera et al. (2020) harness the traces from convolutional layers of generative adversarial networks in fake image detection. Bonomi et al. (2021) analyze spatiotemporal texture dynamics of video signals for Deepfake detection. A plethora of works focus on facial forgery or Deepfake detection using convolution net-based classifiers (Cozzolino et al., 2017; Bayar & Stamm, 2016; Rahmouni et al., 2017; Raja et al., 2017; Zhou et al., 2017; Dogoulis et al., 2023). Rössler et al. (2019) proposed a face forensics dataset and train ResNet (He et al., 2015) and XceptionNet (Chollet, 2016) based classifiers using it. However, as noted in Haliassos et al. (2021), machine learning-based detectors are often vulnerable to novel input perturbations. Such limitations challenge the practical utility of these methods, as any real-world detector should achieve good performance while being robust to small perturbations in the input.

# 3 ROBUSTNESS OF IMAGE WATERMARKING FOR AI-IMAGE DETECTION

In this section, we first present our theoretical results on fundamental constraints for watermarking methods followed by our practical attacks. Proofs are presented in Appendix B.

## 3.1 FUNDAMENTAL CONSTRAINTS FOR WATERMARKING METHODS

Consider $\mathcal{F}$ to represent the distribution of images that have been watermarked using a particular key string $k$, while $\mathcal{R}$ represents the distribution of non-watermarked images.

**Definition 1** (Evasion and Spoofing Errors). *Consider a watermark detector $D$ that predicts values of $0$ and $1$, for non-watermarked and watermarked images, respectively. We define evasion error ($e_0$) and spoofing error ($e_1$) of $D$ on distributions $\mathcal{R}$ and $\mathcal{F}$ as follows:*

$$e_0(\mathcal{F}, D) = \mathbb{P}_{x \sim \mathcal{F}}[D(x) = 0] \quad and \quad e_1(\mathcal{R}, D) = \mathbb{P}_{x \sim \mathcal{R}}[D(x) = 1] \tag{1}$$

We measure distance between the distributions $\mathcal{R}$ and $\mathcal{F}$ using the Wasserstein metric defined as:

$$\mathsf{W}(\mathcal{R}, \mathcal{F}) = \inf_{\gamma \in \Gamma(\mathcal{R}, \mathcal{F})} \mathbb{E}_{(x_1, x_2) \sim \gamma}\big[\|x_1 - x_2\|\big], \tag{2}$$

where $\Gamma(\mathcal{R}, \mathcal{F})$ is the set of all joint probability distributions of $\mathcal{R}$ and $\mathcal{F}$.

**Definition 2.** *(Diffusion Purification) Diffusion purification using a denoising diffusion probabilistic model consists of two steps. In the first step, an image $x_0$ is received and $x_t$ is calculated as:*

$$x_t \sim \mathcal{N}(\sqrt{\bar{\alpha}_t} x_0, (1 - \bar{\alpha}_t) I),$$

*where $\bar{\alpha}_t$ is an increasing function of $t$ that spans from 1 to 0 as $t$ progresses from 0 to 1. Afterward, $x_t$ is denoised using a denoising model to output an image $x_0^{out}$. The denoising model is trained to minimize $\|x_0^{out} - x_0\|$. We represent diffusion purification as $DP_t(.)$ where $x_0^{out} \sim DP_t(x_0)$.*

This technique was previously used in some other applications. For instance, in a prior study (Nie et al., 2022), it was employed to eliminate adversarial perturbations from images as a defense strategy against adversarial attacks. In the following theorem, we claim that applying diffusion purification on images establishes a lower bound on the sum of evasion and spoofing errors of watermark detectors. Luo (2022) presents comprehensive details on denoising diffusion models and their associated parameters, including $\bar{\alpha}_t$.

Let $\mathcal{R}^t$ be the distribution of $x_0^{out} \sim DP_t(x_0)$ where $x_0 \sim \mathcal{R}$. Similarly, define $\mathcal{F}^t$. Below, we provide a lower bound on the detector's error after performing diffusion purification on $\mathcal{R}$ and $\mathcal{F}$.

**Theorem 1.** *The sum of evasion and spoofing errors of a watermark detector $D$ on distributions $\mathcal{R}^t$ and $\mathcal{F}^t$ is lower bounded as follows:*

$$e_0(\mathcal{F}^t, D) + e_1(\mathcal{R}^t, D) \geq 1 - \mathsf{erf}\left(\frac{\sqrt{\bar{\alpha}_t}\, \mathsf{W}(\mathcal{R}, \mathcal{F})}{2\sqrt{2(1 - \bar{\alpha}_t)}}\right),$$

*where $\mathsf{erf}(.)$ is the Gauss error function, and the Wasserstein distance is measured w.r.t the $\ell_2$ norm.*

In Appendix A.2, we elaborate on how this theorem can be extended to apply diffusion purification in the latent space rather than the pixel space. Theorem 1 implies that when the Wasserstein distance between the watermarked and non-watermarked distributions is low (either in pixel or latent spaces), i.e., watermarking with a low perturbation budget, diffusion purification is effective in compromising the watermark. The lower bound presented in Theorem 1, employing real-world configurations of a practical diffusion model, is illustrated in Figure 2, demonstrating the applicability of the theoretical findings in practical scenarios (i.e., the value of error lower bound is considerable, for real-world values of Wasserstein distance).

We note that, even though Theorem 1 is stated w.r.t. using diffusion models as the method to denoise images after adding Gaussian noise to them, our theoretical bound can be attained with any arbitrary denoising technique (Elad et al., 2023; Wang et al., 2022b) (refer to Appendix B for more information). A stronger denoising technique permits the use of a higher magnitude of Gaussian noise, resulting in a more significant lower bound on the error according to Theorem 1.

In the next section, we provide empirical evidence supporting our theoretical result.

### 3.2 Low Perturbation Budget Watermarks: Empirical Attacks

We first categorize certain established watermarking methods into two groups: "low" and "high" perturbation budget watermarks. This categorization relies on the image space $\ell_2$ distance between corresponding watermarked and non-watermarked samples for these methods, as detailed in Table 1. We opt for the $\ell_2$ distance as a surrogate for the actual Wasserstein distance, as it offers an upper bound on the Wasserstein distance, and computing the exact Wasserstein distance is expensive.

In this section, we leverage Theorem 1 to attack watermarking techniques with low perturbation budgets (i.e., known as *imperceptible or invisible watermarks* in prior work) utilizing the diffusion purification attack as outlined in Definition 2. We will discuss attacks on "high" perturbation budget watermarks in the next subsection.

We use 64-bit binary keys for watermarking techniques. Our evaluation is conducted on a set of 100 images drawn from the ImageNet dataset (Russakovsky et al., 2015), and their watermarked counterparts using each method. For the WatermarkDM method, which necessitates pre-training of its models, we undertake training of the injector and detector models for 20 epochs on the ImageNet dataset. Watermark detectors, when given an input image and an encryption key, produce a confidence score that corresponds to the likelihood of the image being watermarked with that specific

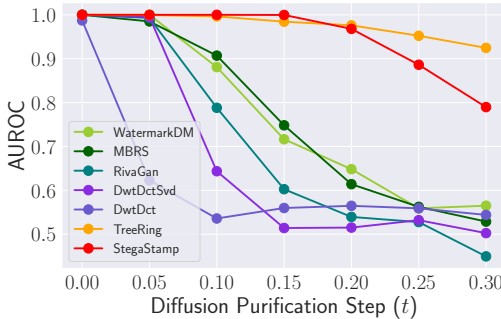
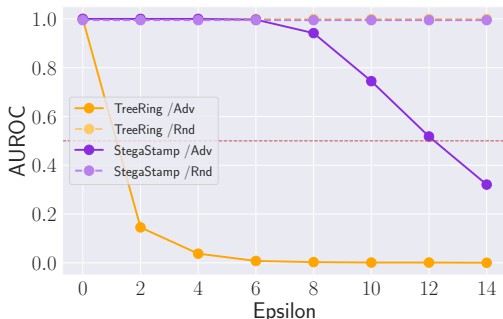

Figure 4: AUROC of watermarking methods against diffusion purification attack for a range of $t$ values. As expected, the robustness of methods against this attack has a correlation with the average image $\ell_2$ distance from Table 1.

Figure 5: AUROC of high-perturbation watermarking methods against $\ell_\infty$ adversarial attack w.r.t adversarial perturbation size $\epsilon$. The colored dashed lines measure robustness against uniform random noise in the range $[-2\epsilon, 2\epsilon]$.

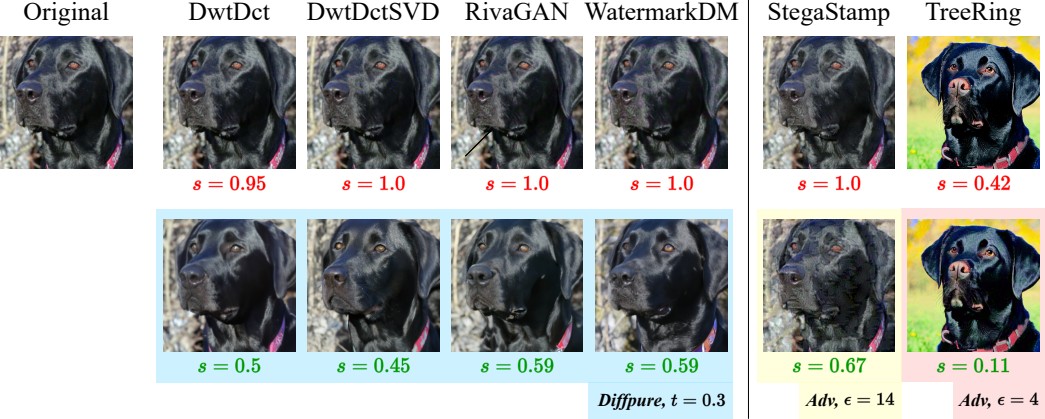

Figure 6: Illustrations of images subjected to the image diffusion purification attack and our adversarial model substitution attack. The $s$ value represents the confidence score assigned by the watermark detector to the images where a higher score indicates a greater likelihood of the image being watermarked. These attacks are able to significantly reduce the AUROC of the detectors (details can be found in Figures 3 and Figure 5.)

key. Subsequently, these images are categorized as watermarked if the confidence score exceeds a predefined threshold, which may either be a constant value or a threshold that varies. In our experiments, we specifically adopt a variable threshold for the process of watermark detection, and use AUROC (Area Under the Receiver Operating Characteristic) measure as our evaluation metric.

The diffusion purification attack, as defined in Definition 2, involves a two-step process: adding noise to images and then denoising them using a denoising model. In a diffusion model with $N$ steps, a diffusion purification attack with parameter $t \in [0, 1]$ on image $x_0$ creates a noisy image $x_t \sim \mathcal{N}(\sqrt{\bar{\alpha}_t} x_0, (1 - \bar{\alpha}_t)I)$ and denoises it with a trained neural network over $N \times t$ steps. Based on Theorem 1, the diffusion purification attack is expected to lower the performance of watermarking methods, particularly when there is a low Wasserstein distance between the distributions of watermarked and non-watermarked images.

We make use of the image diffusion models presented in Nie et al. (2022), particularly a $256 \times 256$ unconditional guided diffusion model designed for ImageNet images. As illustrated in Figure 3, it becomes evident that all the examined watermarking methods can be compromised through a diffusion purification attack with $t = 0.2$. Additionally, we carry out a latent diffusion purification attack, the results of which are detailed in Appendix A.2. Choosing a higher value of $t$ results in a better attack success rate, however, it might degrade the quality of the generated images. Some examples of attacked images using different values of $t$ are shown in Figure 6, and the quality of output images is measured in Table 2 using image quality metric. The diffusion purification

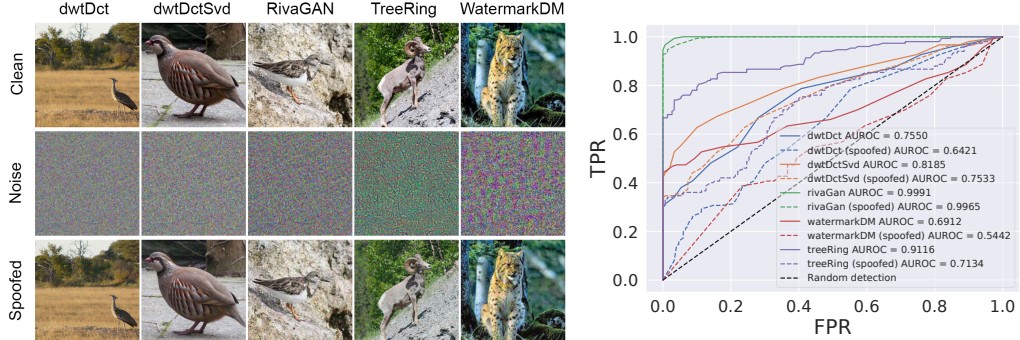

Figure 7: **Left:** The figure demonstrates the spoofing of watermarking techniques, comprising clean ImageNet dataset images (top row), noisy watermarked images (middle row), and spoofed watermarked images (bottom row). Spoofed images blend clean and noisy ones, enabling detection as watermarked. **Right:** Spoofing attack on various watermarking methods.

method lowers AUROC by reducing the detector's confidence in watermarked images and does not consistently boost the confidence of non-watermarked ones. This is understandable as the space of watermarked images is typically much smaller than non-watermarked images due to the key string size. Since diffusion purification is a no-box attack, it cannot apply specific watermark patterns to non-watermarked images without prior knowledge of methods or key strings.

### 3.3 HIGH PERTURBATION BUDGET WATERMARKS: EMPIRICAL ATTACKS

For the watermarking methods that impose high perturbations to the inputs (i.e., TreeRing (Wen et al., 2023) and StegaStamp (Tancik et al., 2020)), our bound in Theorem 1 becomes vacuous since the Wasserstein distance between watermarked and non-watermarked distributions becomes large. In fact, Figure 4 shows empirical evidence that as the perturbation budget of watermarking methods increases, the diffusion purification attack becomes less effective, e.g., TreeRing shows strong robustness against that.

The StegaStamp watermarking (Tancik et al., 2020) imposes large perturbations in the feature space. While its $\ell_2$ perturbation in the image space is larger than that of other invisible watermarking methods (Table 1), it is much smaller than that of the TreeRing. That is partially the reason that the diffusion purification attack in the image space is relatively successful against StegaStamp, especially at higher values of $t$ such as $0.3$ which might leave some artifacts on images. Nevertheless, we categorize StegaStamp as a high perturbation budget watermarking and study additional alternative attacks against it in this section.

For the high perturbation budget watermarking schemes, we develop a model substitution adversarial attack that can successfully alter the watermark detector's decision. To do this, we first train a ResNet-18 (He et al., 2015) classifier on the train split to distinguish between watermarked and non-watermarked images. Then, we target the watermark detector by executing PGD adversarial attacks on test split using the substitute classifier that we have trained. Interestingly, this attack transfers well to the original watermark detector which we assume we do *not* have a white box access to.

Figure 5 displays the AUROC of the methods following adversarial attacks using various adversarial perturbation budgets denoted as $\epsilon$. StegaStamp demonstrates greater resilience to our attack, requiring an $\epsilon$ value of $12/255$ before its performance degrades to the level of a random detector. We note that this level of adversarial noise may leave perceptible artifacts on the images. However, TreeRing is found to be more vulnerable, as a perturbation budget as low as $\epsilon = 2/255$ can render it completely ineffective. Note that the transferability of the adversarial attacks is reliant on the substitute classifier's architecture and training procedure. In our case, we employ a basic ResNet-18 model with standard training procedures. Opting for a more suitable model configuration may lead to a substantial increase in the attack's success rate on the watermark detector. More details about the adversarial attack can be found in Appendix A.3.

### 3.4 SPOOFING ATTACKS ON WATERMARKING METHODS

An effective watermarking method should minimize both spoofing and evasion errors. High spoofing errors enable adversaries to manipulate natural images, leading to a "spoofing attack". Such attacks can falsely identify obscene images as watermarked, potentially harming the reputation of the developers of a watermarked generative model. In this section, we evaluate various watermarking techniques in the presence of adversarial spoofing attempts.

We use a simple strategy to spoof various watermarking techniques by blending watermarked noisy images with clean images (see Algorithm 1). A detailed explanation of the attack is available in Appendix A.4. Figure 7 shows examples of spoofed images for various watermarking methods. While evaluating the AUROC metric, we also augment the images in our dataset using two different techniques: random cropping to 200×200-dimensional images and resizing back to 256×256-dimensional images, and random rotations between -30 and 30 degrees. Figure 7 shows ROC curves for our spoofing attack. As seen here, the AUROC and TPR at low FPR metrics of all the watermarking methods considered here drop after our spoofing attack. RivaGAN seems to be the most robust to our spoofing attack. However, at low FPR regimes, some of the RivaGAN images can be spoofed as well.

## 4 ROBUSTNESS-RELIABILITY TRADE-OFF OF DEEPFAKE DETECTORS

A reliable deepfake detector should exhibit the following two properties: (i) *Robustness:* Minor input image perturbations should not influence performance. (ii) *Reliability:* The detector should accurately identify fake images while minimizing false positives. In this section, we extend the techniques used in proving Theorem 1 to show a fundamental trade-off between these two properties.

Let $\mathcal{R}$ and $\mathcal{F}$ denote the distributions of real and fake images. Consider a detector $D$ that maps an input image $x \in \mathbb{R}^d$ to a latent representation $\phi(x) \in \mathbb{R}^l$ that encodes the perceptual features of the image and uses this representation for detection. We define the robustness of $D$ as its ability to correctly classify a noisy version of the image in this latent space. Let $\mathcal{N}(\phi(x), \sigma)$ denote the distribution of noisy versions of the image $x$ in the latent space, where $\sigma$ is a parameter representing the size of the noise distribution. Here, $\mathcal{N}$ represents a general noise distribution with size parameter $\sigma$. For example, $\mathcal{N}(\phi(x), \sigma)$ could represent an isometric Gaussian distribution with variance $\sigma^2$ or a uniform distribution with width $\sigma$ centered at $\phi(x)$.

**Definition 3** (Robust Detector). *We say a detector $D$ is $(\sigma, \alpha)$-robust on $\mathcal{R}$ and $\mathcal{F}$ under noise distribution $\mathcal{N}$ if, for an image $x$ drawn from either $\mathcal{R}$ or $\mathcal{F}$, its prediction is consistent on latent representations from $\mathcal{N}(\phi(x), \sigma)$ with probability at least $(1 - \alpha)$, for some $\alpha \geq 0$, i.e.,*

$$\forall k \in \{0, 1\}, \forall \mathcal{P} \in \{\mathcal{R}, \mathcal{F}\}, \quad \mathbb{P}_{x \sim \mathcal{P}, \tilde{\phi} \sim \mathcal{N}(\phi(x), \sigma)} \left[ D(\tilde{\phi}) = k | D(\phi(x)) = k \right] \geq 1 - \alpha. \quad (3)$$

This indicates a robust detector's prediction should remain largely unchanged for noisy inputs.

To measure the distance between two distributions $\mathcal{R}$ and $\mathcal{F}$ we use the Wasserstein metric, following a similar formulation as Equation 2. However, here, we define the distance with respect to a norm $\| \cdot \|$ in the latent space $\mathbb{R}^l$ as follows:

$$\mathsf{W}(\mathcal{R}, \mathcal{F}) = \inf_{\gamma \in \Gamma(\mathcal{R}, \mathcal{F})} \mathbb{E}_{(x_1, x_2) \sim \gamma} \left[ \| \phi(x_1) - \phi(x_2) \| \right]. \quad (4)$$

Consider two images $x_1$ and $x_2$. Let $\psi_\sigma(\cdot)$ denote a concave upper bound on the total variation between the corresponding noise distributions $\mathcal{N}(\phi(x_1), \sigma)$ and $\mathcal{N}(\phi(x_2), \sigma)$ as a function of the distance $\| \phi(x_1) - \phi(x_2) \|$ between the corresponding images in the latent space, i.e.,

$$\mathsf{TV}\big(\mathcal{N}(\phi(x_1), \sigma), \mathcal{N}(\phi(x_2), \sigma)\big) \leq \psi_\sigma\big(\| \phi(x_1) - \phi(x_2) \|\big). \quad (5)$$

Note that a concave upper bound like this always exists for any noise distribution $\mathcal{N}$. This is because the total variation between the noise distributions for two images goes from zero to one as the distance between them in the latent space increases. Thus, a trivial bound could be obtained by simply considering the convex hull of the region under the curve of the total variation with respect to the distance. In the case where $\mathcal{N}$ is an isometric Gaussian and the distance is measured using the $\ell_2$ norm, this bound takes the form of the Gauss error function, more precisely:

$$\psi_\sigma\big(\| \phi(x_1) - \phi(x_2) \|_2\big) = \mathsf{erf}\left( \frac{\| \phi(x_1) - \phi(x_2) \|_2}{2\sqrt{2}\sigma} \right).$$

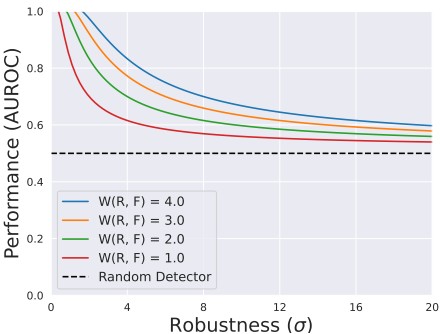 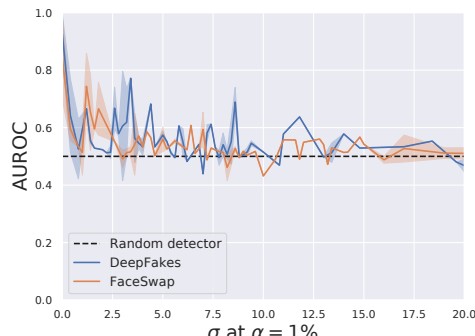

Figure 8: Detection performance w.r.t robustness parameter $\sigma$ for different values of the Wasserstein distance between real $\mathcal{R}$ and fake $\mathcal{F}$ distributions.

Figure 9: AUROC vs. $\sigma$ for a robust deepfake detector with ResNet-18 backbone on DeepFakes (deepfakes) and FaceSwap (MarekKowalski) datasets.

**Theorem 2.** *A $(\sigma, \alpha)$-robust detector's* AUROC *is upper bounded as follows:*

$$\mathsf{AUROC}(D) \leq \frac{1}{1-\alpha}\left(\psi_\sigma(\mathsf{W}_\phi(\mathcal{R},\mathcal{F})) - \frac{\psi_\sigma(\mathsf{W}_\phi(\mathcal{R},\mathcal{F}))^2}{2}\right) + \frac{1+2\alpha-2\alpha^2}{2(1-\alpha)}.$$

For example, when the Wasserstein distance is measured using $\ell_2$ in the latent space and the noise is isometric Gaussian with variance $\sigma^2$, $\psi_\sigma$ takes the form of the Gauss error function: $\psi_\sigma(z) = \mathsf{erf}(z/(2\sqrt{2}\sigma))$. We set $\alpha$ to some small positive value (i.e., $\alpha = 1\%$) and analyze the behavior of the bound for different values of $\sigma$. Figure 8 shows the behavior of the bound with respect to the robustness parameter $\sigma$ for different values of the Wasserstein distance while Figure 16 shows the behavior of the bound with respect to the Wasserstein distance for different values of $\sigma$. The detection performance bound has a negative relationship with the amount of noise that can be tolerated.

**Experiments.** We perform experiments on the images from the FaceForensics++ dataset hosted by Rössler et al. (2019) to verify our theoretical insights empirically. We use ImageNet pretrained ResNet-18 (He et al., 2015) (based on popular DeepFake detectors (Rössler et al., 2019; Dessa, 2019)) and VGG-16-BN (Simonyan & Zisserman, 2014). More details on dataset preprocessing and experiments are provided in Appendix F. We train the models to classify between real and synthetic facial images. The initial trained layers of the models are fixed to be the latent representation $\phi$ given in Equation 3. The remaining layers of the models represent detector $D$. For both ResNet-18 and VGG-16-BN, we choose every layer except the last two convolution layer blocks to represent $\phi$. Detectors with varying robustness to random noise are trained using noisy latent space feature vectors output from $\phi$. We train different detectors with the standard deviation of noise $\sigma$ varied from 0 to 20. For different detectors, we compute the inference $\sigma$ on the test dataset at which they achieve an $\alpha$ of 0.01 using Equation 3. In Appendix F (Figure 18), we show that the detector's robustness (inference $\sigma$ at $\alpha = 1\%$) to random noise increases as the training sigma increases. We use ten randomly sampled Gaussian noises for each sample $\phi(x)$ for this evaluation. After five independent trials, we plot AUROC vs. $\sigma$ for various $(\sigma, \alpha = 0.01)$-robust detectors in Figure 9 using a ResNet-18 backbone for the detector (see plot using VGG-16-BN in Figure 17). Our empirical results show that as the robustness or $\sigma$ at fixed $\alpha$ increases, the AUROC or the performance of the detectors drops.

## 5 Conclusion

In this work, we studied the robustness of AI-image detection methods. We proposed diffusion purification as a certified attack against low-perturbation watermarks, and a model substitution adversarial attack against high-perturbation watermarks. Furthermore, we showed a fundamental reliability vs. robustness trade-off for classifier-based deepfake detectors. Based on our results, designing a robust watermark is a challenging, but not necessarily impossible task. An effective method should possess specific attributes, including a substantial enough watermark perturbation, resistance to naive classification, and resilience to noise transferred from other watermarked images.

## 6 ETHICS STATEMENT

In our research, we follow academic integrity and responsible AI practices. We aim to contribute to discussions on the security of detecting AI-generated content. We prioritize ethical considerations and focus on the societal impact of our findings. Our commitment is to transparency and awareness in the evolving field of generative AI technologies, with the goal of preventing misuse while encouraging progress.

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

| Method | Image $\ell_2$ distance | Latent $\ell_2$ distance |
|---|---|---|
| RivaGAN (Zhang et al., 2019b) | 4.19 | 8.47 |
| DwtDct (Cox et al., 2007) | 5.59 | 5.47 |
| DwtDctSvd (Cox et al., 2007) | 5.54 | 6.67 |
| WatermarkDM (Zhao et al., 2023b) | 7.26 | 13.84 |
| StegaStamp (Tancik et al., 2020) | 17.40 | 118.17 |
| TreeRing (Wen et al., 2023) | 117.58 | 52.81 |

Table 1: Average $\ell_2$ distance between corresponding watermarked and non-watermarked images for each method. The latent representations were obtained using a VQGAN model (Esser et al., 2021), commonly used for latent diffusion models. We consider the first four methods as low perturbation, and the last two as high perturbation ones.

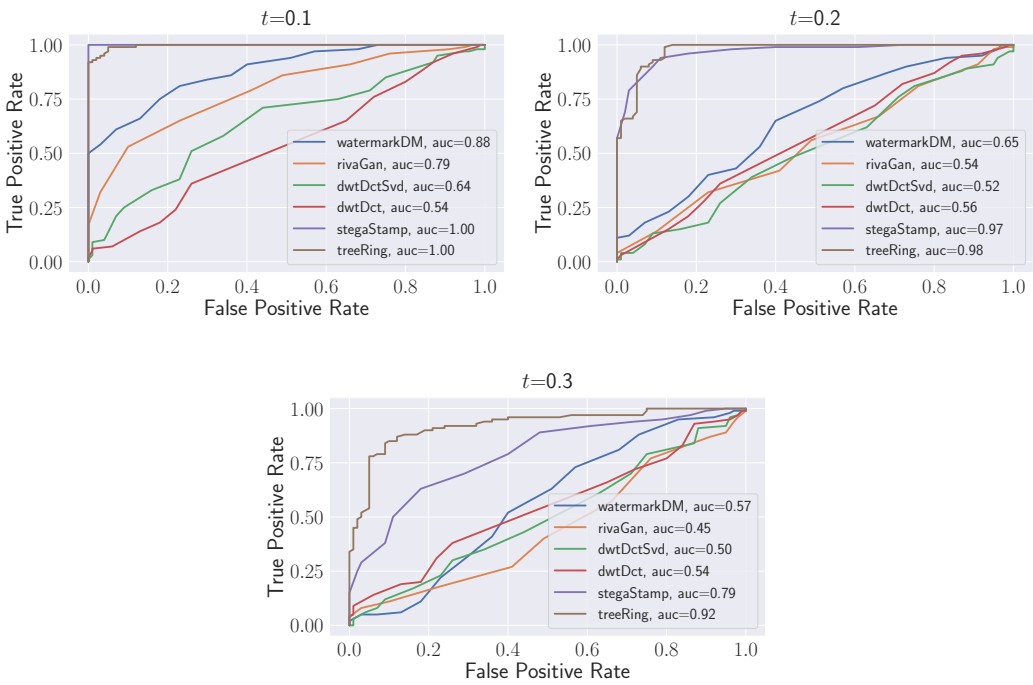

Figure 10: ROC curves for watermarking methods against diffusion purification attack with different values of $t$.

## A COMPLEMENTARY RESULTS FOR WATERMARKING METHODS

### A.1 DIFFUSION PURIFICATION ATTACK

Figure 11 showcases images that have undergone the diffusion purification attack with varying $t$ values, while Figure 10 displays ROC curves for watermarked techniques under these attacks. In the low FPR regime, the TPR of all methods declines at some value of $t$. Table 2 numerically measures the quality of watermarked images that are attacked using diffusion purification w.r.t. the non-attacked images. The quality of images is measured using image quality metrics such as PSNR (Peak Signal-to-Noise Ratio) and SSIM (Structural Similarity Index Measure).

Note that the quality of images for the TreeRing watermark depends on the captions that are provided for the images, and in our case, we are using simple captions based on ImageNet classes. Therefore, the images watermarked by TreeRing might exhibit dissimilarity compared to their non-watermarked counterparts. However, this does not influence the results when attacking the TreeRing

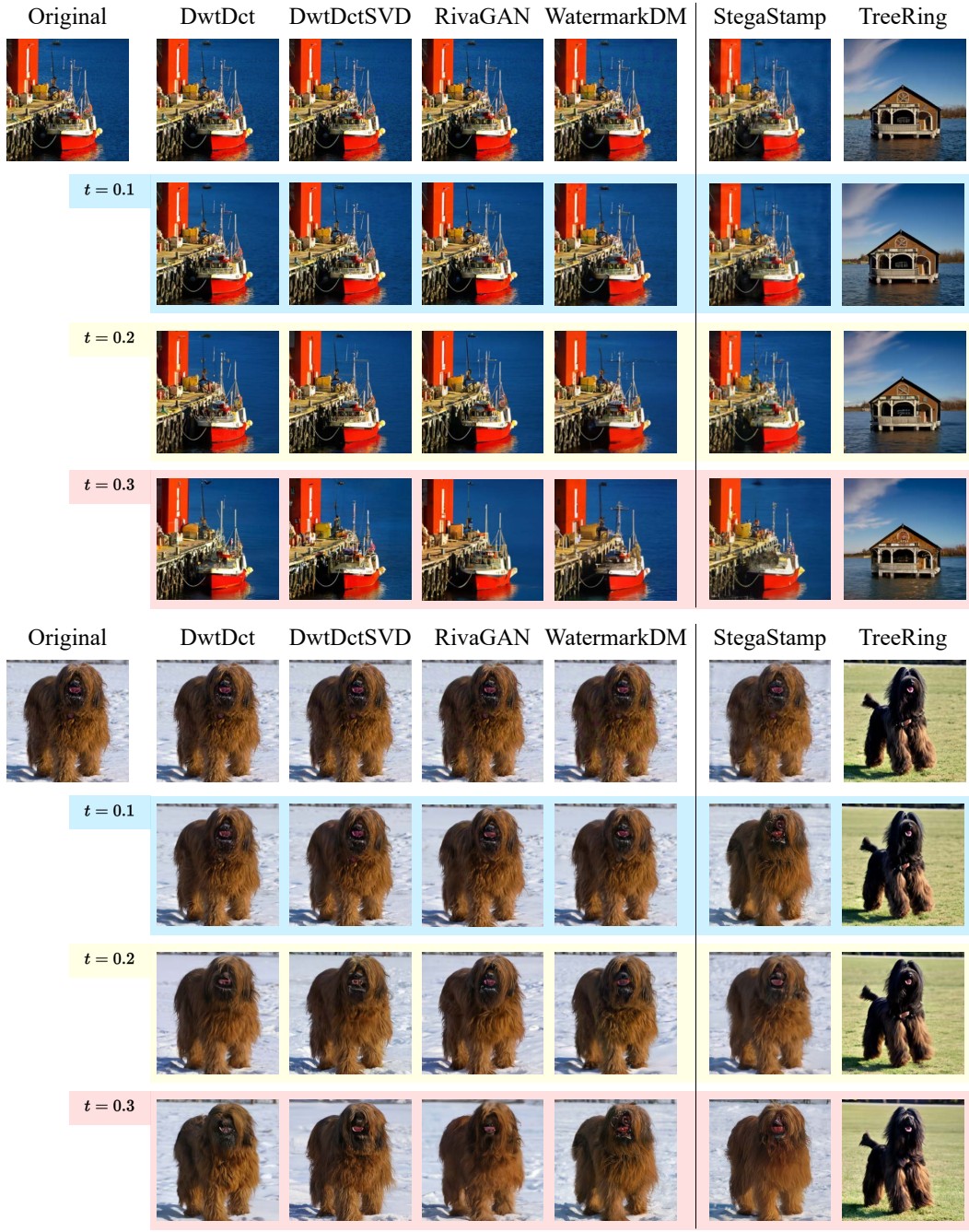

Figure 11: Watermarked images subjected to the image diffusion purification attack are shown with varying values of the parameter $t$. For $t = 0.3$, the attack may excessively alter images, making it unsuitable for some applications.

| Method | PSNR | | | SSIM | | |
|--------|------|------|------|------|------|------|
| | $t = 0.1$ | $t = 0.2$ | $t = 0.3$ | $t = 0.1$ | $t = 0.2$ | $t = 0.3$ |
| RivaGAN | 29.77 | 26.10 | 23.61 | 0.83 | 0.72 | 0.63 |
| DwtDct | 29.64 | 26.03 | 23.70 | 0.83 | 0.72 | 0.63 |
| DwtDctSvd | 29.69 | 26.08 | 23.60 | 0.83 | 0.72 | 0.63 |
| WatermarkDM | 30.33 | 26.41 | 23.87 | 0.86 | 0.75 | 0.66 |
| MBRS | 29.96 | 26.23 | 23.76 | 0.83 | 0.73 | 0.64 |
| StegaStamp | 30.35 | 26.52 | 24.08 | 0.84 | 0.73 | 0.64 |
| TreeRing | 32.45 | 28.27 | 25.49 | 0.92 | 0.86 | 0.81 |

Table 2: Analysis of the quality of images after being attacked using diffusion purification.

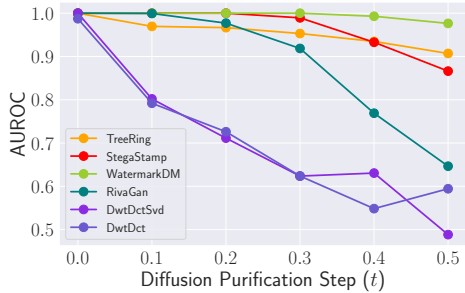 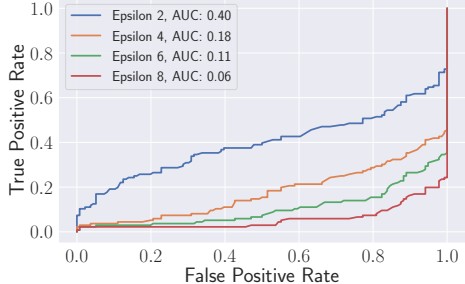

Figure 12: AUROC of watermarking methods against latent diffusion purification attack w.r.t the value of $t$.

Figure 13: ROC curves for attacking watermarked and non-watermarked images that are generated with text from LAION-captions with the TreeRing method.

watermark. Nevertheless, in Fig 13, we demonstrate that our adversarial attacks on TreeRing also extend successfully to captions from LAION-captions data.

## A.2 LATENT DIFFUSION PURIFICATION ATTACK

A similar bound from Theorem 1 can be proven for latent diffusion models. The diffusion process for a latent diffusion model consists of: mapping $x_0$ to the latent space, i.e., $z_0 = \phi(x_0)$; calculating $z_0^{out} \sim DP_t(z_0)$ using a latent diffusion model; and mapping $z_0^{out}$ back to image space, i.e., $x_0^{out} = \phi^{-1}(z_0^{out})$. In this case, since the noise is applied to latent space $\phi$, the Wasserstein distance in Theorem 1 will be replaced by the Wasserstein distance of the latent distributions, i.e., $W(\mathcal{R}_\phi, \mathcal{F}_\phi)$ with $\mathcal{R}_\phi$ being the distribution of images $z_0 = \phi(x_0)$ where $x_0 \sim \mathcal{R}$, and $\mathcal{F}_\phi$ defined similarly.

In practice, we perform latent diffusion purification attack by employing a Text-Guided Image-to-Image Stable Diffusion model (Rombach et al., 2022), and using BLIP model (Li et al., 2022) to generate image captions, as guidance for diffusion models. Figure 12 includes the AUROC of watermarking methods against this attack, and Figure 14 contains samples output images for this attack.

## A.3 ADVERSARIAL ATTACK

We conduct adversarial attacks involving model substitution on high-perturbation budget watermarks, specifically StegaStamp and TreeRing. Our training dataset comprises $7,500$ watermarked and $7,500$ non-watermarked images. For StegaStamp, we use images sourced from ImageNet, along with their watermarked versions, for both training and testing. In contrast, for TreeRing, the non-watermarked images can either be sourced from ImageNet or generated using a process similar to TreeRing's watermarking method, but employing random noise instead of TreeRing's key string. We have observed through empirical testing that the effectiveness of our adversarial attack remains consistent, regardless of the choice between these two types of non-watermarked training data. As a result, we opted for the latter approach.

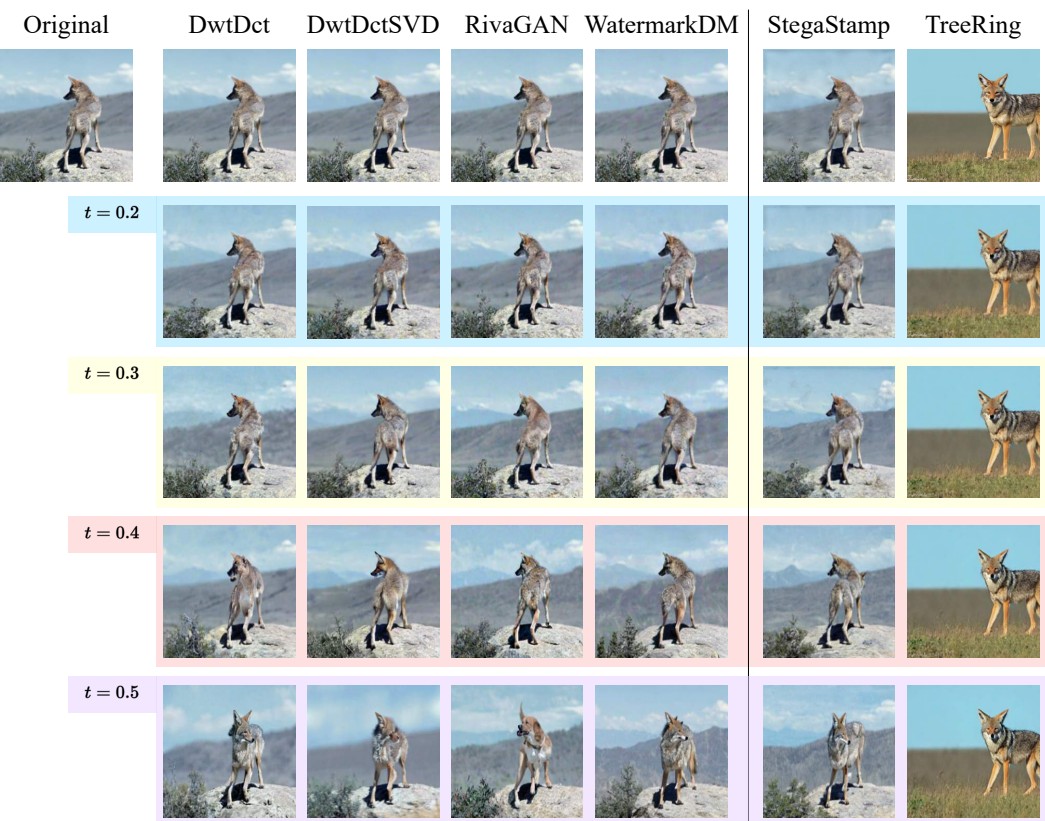

Figure 14: Watermarked Images subjected to the latent diffusion purification attack are shown with varying values of the parameter $t$. For $t = 0.5$, the attack drastically changes the images in most cases (except for TreeRing).

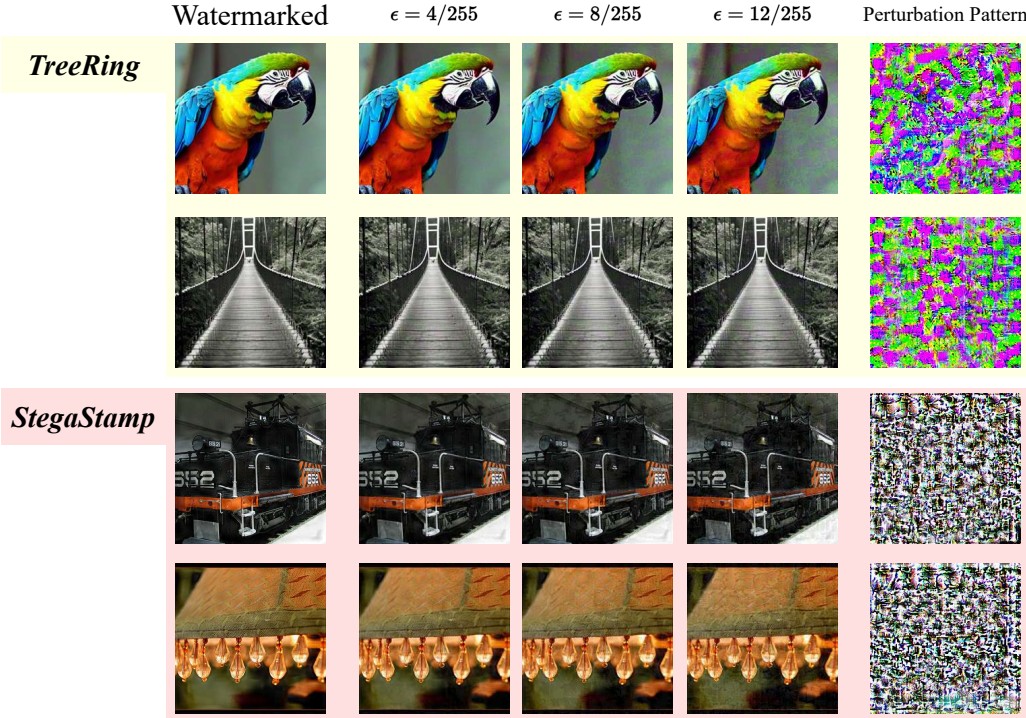

Figure 15: Watermarked images subjected to the model substitution adversarial attack are shown with varying values of adversarial perturbation budget $\epsilon$. Attacks on images watermarked with the same method show similar perturbation patterns.

For StegaStamp, we employ 100-bit binary keys, mirroring the key length described in their report. In the case of TreeRing, we stick to the ring-type key employed in the original implementation. TreeRing necessitates captions for generating watermark images, and for our ImageNet data, we utilize captions structured as "a photo of a ⟨imagenet-class⟩." Nevertheless, in Figure 13, we demonstrate that our attacks on TreeRing also extend successfully to LAION-captions data (Schuhmann et al., 2021).

Our substitute classifiers are trained for 10 epochs and receive higher than $99.8\%$ accuracy on validation data. For StegaStamp, we observed that augmenting the training data with Gaussian noise improves the transferability of the attacks on the watermark detector.

To launch adversarial attacks on images using substitute classifiers, we employ a PGD attack with 300 iterations and a step size denoted as $\alpha = 0.05\epsilon$. Our observations indicate that adversarial perturbations for a particular watermarking method exhibit a roughly consistent pattern. Consequently, we initiate our adversarial attacks on each image from the perturbation discovered for the previous image, a technique akin to the one employed in Shafahi et al. (2019). To ensure the accurate identification of the perturbation pattern, we execute a series of preliminary warm-up attacks at the outset. Some sample adversarial images can be seen in Figure 15.

In Figure 13, we present ROC curves for attacking TreeRing images that are generated with text from LAION-captions data (Schuhmann et al., 2021). This shows that our adversarial attack which is performed on the classifier trained on ImageNet data, generalizes to any images watermarked using the TreeRing method.

## A.4    SPOOFING

To perform the spoofing attack, we first generate random noisy images where pixels are drawn from different Gaussian distributions with varying standard deviations. The noisy images are normalized

| Method | | Base | Blur ($k = 5$) | JPEG | DiffPure ($t = 0.2$) |
|---|---|---|---|---|---|
| RivaGAN | $t = 0.1$ | 0.655 | **0.692** | 0.679 | 0.638 |
| | $t = 0.2$ | **0.623** | 0.607 | 0.604 | 0.593 |
| | $t = 0.3$ | **0.579** | 0.568 | 0.562 | 0.555 |
| DwtDct | $t = 0.1$ | **0.548** | 0.546 | 0.544 | 0.542 |
| | $t = 0.2$ | **0.542** | 0.540 | 0.539 | 0.539 |
| | $t = 0.3$ | **0.539** | 0.538 | 0.538 | 0.537 |
| DwtDctSvd | $t = 0.1$ | 0.560 | 0.566 | **0.567** | **0.567** |
| | $t = 0.2$ | **0.564** | 0.560 | 0.561 | 0.558 |
| | $t = 0.3$ | **0.555** | 0.553 | 0.551 | 0.549 |
| WatermarkDM | $t = 0.1$ | 0.876 | **0.885** | 0.805 | 0.597 |
| | $t = 0.2$ | **0.644** | 0.630 | 0.604 | 0.518 |
| | $t = 0.3$ | 0.568 | **0.604** | 0.565 | 0.564 |
| MBRS | $t = 0.1$ | **0.914** | 0.828 | 0.874 | 0.597 |
| | $t = 0.2$ | 0.614 | **0.636** | 0.634 | 0.545 |
| | $t = 0.3$ | 0.536 | 0.493 | 0.444 | **0.547** |
| StegaStamp | $t = 0.1$ | **1.000** | 1.000 | 0.998 | 0.920 |
| | $t = 0.2$ | 0.966 | 0.960 | **0.971** | 0.832 |
| | $t = 0.3$ | 0.781 | **0.802** | 0.767 | 0.659 |
| TreeRing | $t = 0.1$ | **0.996** | 0.989 | 0.947 | 0.935 |
| | $t = 0.2$ | **0.976** | 0.956 | 0.923 | 0.912 |
| | $t = 0.3$ | **0.928** | 0.907 | 0.871 | 0.876 |

Table 3: The AUROC of watermarking methods against diffusion purification attack, after applying post-attack mitigations to the attacked images.

to have values between 0 and 1. For every watermarking method that we evaluate, we apply their watermarks on these noisy images to obtain corresponding watermarked noisy images.

We use an input prompt, "a noisy image", along with the noisy images to generate noisy watermarked TreeRing (Wen et al., 2023) images. Once we obtain the watermarked noisy images, we do a mixup (or image blending) by adding noisy images to the clean images to spoof them. We observe that the watermark signatures in the noisy images help detect the resulting blended images as watermarked.

We provide the pseudocode for spoofing watermarks in Algorithm 1.

---

**Algorithm 1** Watermark Spoofing

---

**Require:** clean image $x$, watermarking model $\mathcal{W}$, mixup parameter $\alpha$
  $z = \text{random}(x.\text{shape})$                    ▷ generate random noise with shape of image $x$
  $z = z - z.\min()$                                  ▷ normalize $z$
  $z = z/z.\max()$
  $z = \alpha\mathcal{W}(z)$          ▷ watermark the noise; for TreeRing, condition with text "a noisy image"
  $\gamma = 1 - z.\max()$
  $x = \gamma x/x.\max()$                       ▷ $z + x$ can now only have a value of maximum 1
  **return** $x + z$                                         ▷ spoofed image

---

### A.5   ROBUSTNESS OF ATTACKS AGAINST MITIGATIONS

In this section, we measure the robustness of the diffusion purification and the model substitution adversarial attacks on image watermarking techniques. This robustness is measured by applying post-attack mitigations such as Gaussian Blur and JPEG Compression to the attacked images. A robust attack is expected to result in a low AUROC on the watermark detector, even after the post-attack mitigations are applied.

| Method | | Base | Blur ($k = 5$) | Blur ($k = 15$) | JPEG | DiffPure ($t = 0.2$) |
|---|---|---|---|---|---|---|
| StegaStamp | $\epsilon = 4$ | **1.000** | 1.000 | 0.999 | 0.991 | 0.879 |
| | $\epsilon = 8$ | **0.923** | 0.838 | 0.791 | 0.864 | 0.703 |
| | $\epsilon = 12$ | 0.492 | 0.424 | 0.341 | 0.496 | **0.566** |
| TreeRing | $\epsilon = 4$ | 0.035 | 0.025 | 0.023 | 0.046 | **0.891** |
| | $\epsilon = 8$ | 0.002 | 0.001 | 0.001 | 0.006 | **0.531** |
| | $\epsilon = 12$ | 0.001 | 0.0002 | 0.0002 | 0.001 | **0.074** |

Table 4: The AUROC of watermarking methods against model substitution adversarial attack, after applying post-attack mitigations to the attacked images.

Table 3 showcases the AUROC of watermarking methods against diffusion purification attacks, after applying post-attack mitigations. The application of post-attack mitigations is not causing significant increases in the AUROC. This is anticipated since the primary aim of the diffusion purification attack is the removal of watermarks from the watermarked images (i.e., to achieve a bit-accuracy close to $0.5$ for both watermarked and non-watermarked images). Therefore, it is reasonable to expect that basic no-box post-attack mitigations will encounter challenges in recovering the watermark.

On the other hand, our proposed adversarial attack has black-box information about the watermark, and therefore, is able to target both non-watermarked and watermarked images for its attack, in order to increase or reduce their watermark bit-accuracy, respectively. Table 4 showcases the AUROC of watermarking methods against the adversarial attack, after applying post-attack mitigations. While post-attack mitigations, specifically DiffPure, are able to increase the AUROC in some cases, they fail to negate the effect of the attack for higher attack budgets such as $\epsilon = 8/255$.

# B PROOF OF THEOREM 1

**Statement.** *The sum of evasion and spoofing errors of a watermark detector $D$ on distributions $\mathcal{R}^t$ and $\mathcal{F}^t$ is lower bounded as follows:*

$$e_0(\mathcal{F}^t, D) + e_1(\mathcal{R}^t, D) \geq 1 - \mathrm{erf}\left(\frac{\sqrt{\bar{\alpha}_t}\, \mathsf{W}(\mathcal{R}, \mathcal{F})}{2\sqrt{2(1 - \bar{\alpha}_t)}}\right).$$

*Proof.* Let $\psi_\sigma(\cdot)$ denote a concave upper bound on the total variation between two noise distributions $\mathcal{N}(x_1, \sigma)$ and $\mathcal{N}(x_2, \sigma)$ as a function of the distance $\|x_1 - x_2\|$ between the corresponding images, i.e.,

$$\mathsf{TV}\big(\mathcal{N}(x_1, \sigma), \mathcal{N}(x_2, \sigma)\big) \leq \psi_\sigma\big(\|x_1 - x_2\|\big), \tag{6}$$

where TV is the total variation of two distributions.

Note that a concave upper bound like this always exists for any noise distribution $\mathcal{N}$. This is because the total variation of the noise distributions for two images goes from zero to one as the distance between them in the latent space increases. Thus a trivial bound could be obtained by simply considering the convex hull of the region under the curve of the total variation with respect to the distance. In the case where $\mathcal{N}$ is an isometric Gaussian and the distance is measured using the $\ell_2$-norm, this bound takes the form of the Gauss error function, more precisely:

$$\psi_\sigma(\|x_1 - x_2\|) = \mathrm{erf}\left(\frac{\|x_1 - x_2\|}{2\sqrt{2}\sigma}\right) \tag{7}$$

Now, consider the distribution of images under the noise distribution $\mathcal{N}$. Let $\mathcal{R}_\mathcal{N}$ be the distribution of images $\tilde{x} \sim \mathcal{N}(x, \sigma)$ where $x \sim \mathcal{R}$. Similarly, define $\mathcal{F}_\mathcal{N}$. The same equality as Equation 7 can be written for the Wasserstein distance of $\mathcal{R}$ and $\mathcal{F}$ defined with respect to $\ell_2$ norm, when $x_1$ and $x_2$ are sampled from $\mathcal{R}$ and $\mathcal{F}$, respectively.

$$\psi_\sigma\big(\mathsf{W}(\mathcal{R}, \mathcal{F})\big) = \mathrm{erf}\left(\frac{\mathsf{W}(\mathcal{R}, \mathcal{F})}{2\sqrt{2}\sigma}\right). \tag{8}$$

We bound the total variation of the noisy distributions $\mathcal{R}_\mathcal{N}$ and $\mathcal{F}_\mathcal{N}$ in terms of the Wasserstein distance between the original distributions $\mathcal{R}$ and $\mathcal{F}$. The reason why this bound holds is that as $\mathcal{R}$ and $\mathcal{F}$ get closer to each other, $\mathcal{R}_\mathcal{N}$ and $\mathcal{F}_\mathcal{N}$ start to overlap due to the noise distribution $\mathcal{N}$ around them.

**Lemma 1.** *The total variation of $\mathcal{R}_\mathcal{N}$ and $\mathcal{F}_\mathcal{N}$, and hence, the success rate of any detector $D$ on these distributions, is upper bounded by a function of the Wasserstein distance of the original distributions $\mathcal{R}$ and $\mathcal{F}$ as follows:*

$$1 - (e_0(\mathcal{F}_\mathcal{N}, D) + e_1(\mathcal{R}_\mathcal{N}, D)) \leq \mathsf{TV}(\mathcal{R}_\mathcal{N}, \mathcal{F}_\mathcal{N}) \leq \psi_\sigma\big(\mathsf{W}(\mathcal{R}, \mathcal{F})\big).$$

*Proof.* For simplicity of the proof, assume $D$ to be deterministic, however, the proof can be generalized for randomized detectors too. Define $E_D = \{x : D(x) = 1\}$. Based on the definition of total variation,

$$\begin{aligned}
\mathsf{TV}(\mathcal{R}_\mathcal{N}, \mathcal{F}_\mathcal{N}) &= \sup_E \left| \mathbb{P}_{\tilde{x}_1 \sim \mathcal{R}_\mathcal{N}}[\tilde{x}_1 \in E] - \mathbb{P}_{\tilde{x}_2 \sim \mathcal{F}_\mathcal{N}}[\tilde{x}_2 \in E] \right| \\
&\geq \left| \mathbb{P}_{\tilde{x}_1 \sim \mathcal{R}_\mathcal{N}}[\tilde{x}_1 \in E_D] - \mathbb{P}_{\tilde{x}_2 \sim \mathcal{F}_\mathcal{N}}[\tilde{x}_2 \in E_D] \right| \\
&= \left| e_1(\mathcal{R}_\mathcal{N}, D) - \big(1 - e_0(\mathcal{F}_\mathcal{N}, D)\big) \right| \qquad \text{(Definition 1)} \\
&\geq 1 - (e_0(\mathcal{F}_\mathcal{N}, D) + e_1(\mathcal{R}_\mathcal{N}, D)).
\end{aligned}$$

Furthermore, the inequality $\mathsf{TV}(\mathcal{R}_\mathcal{N}, \mathcal{F}_\mathcal{N}) \leq \psi_\sigma\big(\mathsf{W}(\mathcal{R}, \mathcal{F})\big)$ can be derived from the proof presented for Lemma 3 in Appendix E, by substituting the latent function $\phi$ with the identity function.

$\square$

In Lemma 1, we have shown that after applying Gaussian noise to $\mathcal{R}$ and $\mathcal{F}$, they become more indistinguishable. However, using Gaussian noise as an attack against image watermarks will degrade the quality of images. Therefore, we utilize denoising diffusion models to remove the added noise. Since the bound in Lemma 1 is on total variation, by applying a denoising function on the noisy distributions $\mathcal{R}_\mathcal{N}$ and $\mathcal{F}_\mathcal{N}$, the bound still holds. Note that our theoretical results do not rely on the utilization of denoising diffusion models, and any arbitrary denoising technique (Elad et al., 2023; Wang et al., 2022b), can be used to achieve similar bounds.

Let $\mathcal{R}_\mathcal{N}^t$ be the distribution of $x_t \sim \mathcal{N}(\sqrt{\bar{\alpha}_t}x_0, (1 - \bar{\alpha}_t)I)$ where $x_0 \sim \mathcal{R}$, and define $\mathcal{F}_\mathcal{N}^t$ similarly. Additionally, define $G^t(.)$ as the function that performs denoising process to $\mathcal{R}_\mathcal{N}^t$ and $\mathcal{F}_\mathcal{N}^t$ (i.e., samples of $\mathcal{R}^t$ come from $x_0^{out} \sim G^t(x_t)$ where $x_t \sim \mathcal{R}_\mathcal{N}^t$).

We use Lemma 1, to get an upper bound on the total variation of $\mathcal{R}_\mathcal{N}^t$ and $\mathcal{F}_\mathcal{N}^t$, with $\sigma = \sqrt{(1 - \bar{\alpha}_t)}$, based on the definition of $\mathcal{R}_\mathcal{N}^t$ and $\mathcal{F}_\mathcal{N}^t$:

$$\begin{aligned}
\mathsf{TV}(\mathcal{R}_\mathcal{N}^t, \mathcal{F}_\mathcal{N}^t) &\leq \psi_\sigma\big(\mathsf{W}(\mathcal{R}, \mathcal{F})\big) \\
&= \mathsf{erf}\big(\frac{\sqrt{\bar{\alpha}_t}\,\mathsf{W}(\mathcal{R}, \mathcal{F})}{2\sqrt{2(1 - \bar{\alpha}_t)}}\big). \qquad \text{(Equation 8)}
\end{aligned}$$

Next, we use the fact that after applying the function $G^t(.)$ on samples from $\mathcal{R}_\mathcal{N}^t$ and $\mathcal{F}_\mathcal{N}^t$, the total variation does not increase, i.e.

$$\mathsf{TV}(\mathcal{R}^t, \mathcal{F}^t) \leq \mathsf{TV}(\mathcal{R}_\mathcal{N}^t, \mathcal{F}_\mathcal{N}^t). \qquad (9)$$

Now, the theorem's statement can be proven as follows:

$$\mathsf{TV}(\mathcal{R}^t, \mathcal{F}^t) \leq \mathsf{TV}(\mathcal{R}_\mathcal{N}^t, \mathcal{F}_\mathcal{N}^t) \leq \mathsf{erf}\big(\frac{\sqrt{\bar{\alpha}_t}\,\mathsf{W}(\mathcal{R}, \mathcal{F})}{2\sqrt{2(1 - \bar{\alpha}_t)}}\big)$$

$$1 - (e_0(\mathcal{F}^t, D) + e_1(\mathcal{R}^t, D)) \leq \mathsf{erf}\big(\frac{\sqrt{\bar{\alpha}_t}\,\mathsf{W}(\mathcal{R}, \mathcal{F})}{2\sqrt{2(1 - \bar{\alpha}_t)}}\big) \qquad \text{(Lemma 1)}$$

$$e_0(\mathcal{F}^t, D) + e_1(\mathcal{R}^t, D) \geq 1 - \mathsf{erf}\big(\frac{\sqrt{\bar{\alpha}_t}\,\mathsf{W}(\mathcal{R}, \mathcal{F})}{2\sqrt{2(1 - \bar{\alpha}_t)}}\big).$$

We note that inequality 9 can be written for any arbitrary denoising function that receives noisy images of $R_{\mathcal{N}}^t$ and $F_{\mathcal{N}}^t$ as inputs, and outputs denoised images with acceptable image quality.

$\square$

## C   PROOF OF THEOREM 2

**Statement.** *The performance of a $(\sigma, \alpha)$-robust detector measured using its* AUROC *is upper bounded as follows:*

$$\mathsf{AUROC}(D) \leq \frac{1}{1-\alpha} \left( \psi_\sigma(\mathsf{W}_\phi(\mathcal{R}, \mathcal{F})) - \frac{\psi_\sigma(\mathsf{W}_\phi(\mathcal{R}, \mathcal{F}))^2}{2} \right) + \frac{1 + 2\alpha - 2\alpha^2}{2(1-\alpha)},$$

*Proof.* We quantify the dissimilarity between the distributions $\mathcal{R}$ and $\mathcal{F}$ using the Wasserstein metric defined with respect to a norm $\|\cdot\|$ in the latent space $\mathbb{R}^l$ as follows:

$$\mathsf{W}_\phi(\mathcal{R}, \mathcal{F}) = \inf_{\gamma \in \Gamma(\mathcal{R}, \mathcal{F})} \mathbb{E}_{(x_1, x_2) \sim \gamma} \big[ \|\phi(x_1) - \phi(x_2)\| \big], \tag{10}$$

where $\Gamma(\mathcal{R}, \mathcal{F})$ is the set of all joint probability distributions of $\mathcal{R}$ and $\mathcal{F}$, i.e.,

$$\Gamma(\mathcal{R}, \mathcal{F}) = \left\{ \gamma : \mathbb{R}^d \times \mathbb{R}^d \to \mathbb{R}_{\geq 0} \,\bigg|\, \int_{\mathbb{R}^d} \gamma(x_1, x_2) dx_2 = \mathsf{pdf}_{\mathcal{R}}(x_1) \right.$$

$$\left. \text{and} \int_{\mathbb{R}^d} \gamma(x_1, x_2) dx_1 = \mathsf{pdf}_{\mathcal{F}}(x_2) \right\},$$

where $\mathsf{pdf}_{\mathcal{R}}$ and $\mathsf{pdf}_{\mathcal{F}}$ represent the probability density functions of $\mathcal{R}$ and $\mathcal{F}$. For the sake of simplicity, we assume that there exists an element $\gamma^* \in \Gamma$ that achieves the infimum. Otherwise, one can derive our results for some $\gamma^*$ that achieves an expected distance of $\mathsf{W}_\phi(\mathcal{R}, \mathcal{F}) + \delta$ for an arbitrarily small $\delta > 0$.

We use the notation $\psi_\sigma(\cdot)$ to represent a concave upper bound on the total variation between two noise distributions, specifically $\mathcal{N}(\phi(x_1), \sigma)$ and $\mathcal{N}(\phi(x_2), \sigma)$. This upper bound is expressed as a function of the distance $\|\phi(x_1) - \phi(x_2)\|$ between the respective images in the latent space, i.e.,

$$\mathsf{TV}\big( \mathcal{N}(\phi(x_1), \sigma), \mathcal{N}(\phi(x_2), \sigma) \big) \leq \psi_\sigma\big( \|\phi(x_1) - \phi(x_2)\| \big). \tag{11}$$

In the case where $\mathcal{N}$ is an isometric Gaussian and the distance is measured using the $\ell_2$-norm, this bound takes the form of the Gauss error function, more precisely:

$$\psi_\sigma\big( \|\phi(x_1) - \phi(x_2)\|_2 \big) = \mathsf{erf}\left( \frac{\|\phi(x_1) - \phi(x_2)\|_2}{2\sqrt{2}\sigma} \right).$$

Now, consider the distribution of noisy real images in the latent space under the noise distribution $\mathcal{N}$. Let $\mathcal{R}_{\mathcal{N}}^\phi$ be the distribution of latent representations $\tilde{\phi} \sim \mathcal{N}(\phi(x), \sigma)$ where $x \sim \mathcal{R}$. Similarly, define $\mathcal{F}_{\mathcal{N}}^\phi$. The following lemma relates the performance of a $(\sigma, \alpha)$-robust detector $D$ under the original and noisy versions of the two distributions.

**Lemma 2.** *The* AUROC *of a $(\sigma, \alpha)$-robust detector $D$ on the original distributions $\mathcal{R}$ and $\mathcal{F}$ is bounded by that for the noisy versions of the distributions $\mathcal{R}_{\mathcal{N}}^\phi$ and $\mathcal{F}_{\mathcal{N}}^\phi$ as follows:*

$$\mathsf{AUROC}(D) \leq \frac{\mathsf{AUROC}_{\mathcal{N}}(D)}{1-\alpha} + \alpha.$$

Proof is available in Appendix D.

Next, we bound the total variation between the noisy distributions $\mathcal{R}_{\mathcal{N}}^\phi$ and $\mathcal{F}_{\mathcal{N}}^\phi$ in terms of the Wasserstein distance between the original distributions $\mathcal{R}$ and $\mathcal{F}$. The reason why this bound holds is that as $\mathcal{R}$ and $\mathcal{F}$ get closer to each other in the latent space, $\mathcal{R}_{\mathcal{N}}^\phi$ and $\mathcal{F}_{\mathcal{N}}^\phi$ start to overlap due to the noise distribution $\mathcal{N}$ around them.

**Lemma 3.** *The total variation between the noisy distributions $\mathcal{R}_\mathcal{N}^\phi$ and $\mathcal{F}_\mathcal{N}^\phi$ is bounded by the Wasserstein distance of the original distributions $\mathcal{R}$ and $\mathcal{F}$ as follows:*

$$\mathsf{TV}(\mathcal{R}_\mathcal{N}^\phi, \mathcal{F}_\mathcal{N}^\phi) \le \psi_\sigma\big(\mathsf{W}_\phi(\mathcal{R}, \mathcal{F})\big).$$

Proof is available in Appendix E.

Now, we use the above two lemmas to put a bound on the performance of the detector on $\mathcal{R}$ and $\mathcal{F}$. We first show that the performance on the noisy distributions $\mathcal{R}_\mathcal{N}^\phi$ and $\mathcal{F}_\mathcal{N}^\phi$ is bounded by the total variation between these distributions. We then use Lemma 3 to convert this total variation distance to the Wasserstein distance between the original distributions $\mathcal{R}$ and $\mathcal{F}$. Finally, we relate the bound to the detector's performance on the original distributions using Lemma 2.

The true positive rate $\mathsf{TPR}_\mathcal{N}$ and the false positive rate $\mathsf{FPR}_\mathcal{N}$ of the detector on the noisy distributions $\mathcal{R}_\mathcal{N}^\phi$ and $\mathcal{F}_\mathcal{N}^\phi$ can be bounded by the total variation between these distributions as follows:

$$
\begin{aligned}
|\mathsf{TPR}_\mathcal{N} - \mathsf{FPR}_\mathcal{N}| &= |\mathbb{P}_{x\sim\mathcal{F},\tilde\phi\sim\mathcal{N}(\phi(x),\sigma)}[D(\tilde\phi) = 1] - \mathbb{P}_{x\sim\mathcal{R},\tilde\phi\sim\mathcal{N}(\phi(x),\sigma)}[D(\tilde\phi) = 1]| \\
&= \mathsf{TV}(\mathcal{R}_\mathcal{N}^\phi, \mathcal{F}_\mathcal{N}^\phi)
\end{aligned}
$$

Since the true positive rate is also bounded by one, we have:

$$\mathsf{TPR}_\mathcal{N} \le \min(\mathsf{FPR}_\mathcal{N} + \mathsf{TV}(\mathcal{R}_\mathcal{N}^\phi, \mathcal{F}_\mathcal{N}^\phi), 1).$$

Denoting $\mathsf{FPR}_\mathcal{N}$, $\mathsf{TPR}_\mathcal{N}$ and $\mathsf{TV}(\mathcal{R}_\mathcal{N}^\phi, \mathcal{F}_\mathcal{N}^\phi)$ with $x, y$, and $tv$, respectively, for brevity, we bound the $\mathsf{AUROC}_\mathcal{N}$ as follows:

$$
\begin{aligned}
\mathsf{AUROC}_\mathcal{N}(D) = \int_0^1 y\,dx &\le \int_0^1 \min(x + tv, 1)\,dx \\
&= \int_0^{1-tv} (x + tv)\,dx + \int_{1-tv}^1 dx \\
&= \left|\frac{x^2}{2} + tvx\right|_0^{1-tv} + |x|_{1-tv}^1 \\
&= \frac{(1 - tv)^2}{2} + tv(1 - tv) + tv \\
&= \frac{1}{2} + \frac{tv^2}{2} - tv + tv - tv^2 + tv \\
&= \frac{1}{2} + tv - \frac{tv^2}{2}.
\end{aligned}
$$

Thus,

$$
\begin{aligned}
\mathsf{AUROC}_\mathcal{N}(D) &= \frac{1}{2} + \mathsf{TV}(\mathcal{R}_\mathcal{N}^\phi, \mathcal{F}_\mathcal{N}^\phi) - \frac{\mathsf{TV}(\mathcal{R}_\mathcal{N}^\phi, \mathcal{F}_\mathcal{N}^\phi)^2}{2} \\
&\le \frac{1}{2} + \psi_\sigma\big(\mathsf{W}_\phi(\mathcal{R}, \mathcal{F})\big) - \frac{\psi_\sigma\big(\mathsf{W}_\phi(\mathcal{R}, \mathcal{F})\big)^2}{2}.
\end{aligned}
$$

(from Lemma 3 and since $1/2 + x - x^2/2$ is increasing in $[0, 1]$)

Finally, from Lemma 2, we have:

$$
\begin{aligned}
\mathsf{AUROC}(D) &\le \frac{\mathsf{AUROC}_\mathcal{N}(D)}{1 - \alpha} + \alpha \\
&\le \frac{1}{1 - \alpha}\left(\frac{1}{2} + \psi_\sigma\big(\mathsf{W}_\phi(\mathcal{R}, \mathcal{F})\big) - \frac{\psi_\sigma\big(\mathsf{W}_\phi(\mathcal{R}, \mathcal{F})\big)^2}{2}\right) + \alpha \qquad \text{(from above)} \\
&= \frac{1}{1 - \alpha}\left(\psi_\sigma(\mathsf{W}(\mathcal{R}, \mathcal{F})) - \frac{\psi_\sigma(\mathsf{W}(\mathcal{R}, \mathcal{F}))^2}{2}\right) + \frac{1 + 2\alpha - 2\alpha^2}{2(1 - \alpha)}.
\end{aligned}
$$

$\square$

# D    PROOF OF LEMMA 2

**Statement.** *The* AUROC *of a* $(\sigma, \alpha)$*-robust detector* $D$ *on the original distributions* $\mathcal{R}$ *and* $\mathcal{F}$ *is bounded by that for the noisy versions of the distributions* $\mathcal{R}_{\mathcal{N}}^{\phi}$ *and* $\mathcal{F}_{\mathcal{N}}^{\phi}$ *as follows:*

$$\mathsf{AUROC}(D) \leq \frac{\mathsf{AUROC}_{\mathcal{N}}(D)}{1 - \alpha} + \alpha.$$

*Proof.* Let $\mathsf{TPR}, \mathsf{FPR}, \mathsf{TPR}_{\mathcal{N}}$ and $\mathsf{FPR}_{\mathcal{N}}$ denote the true and false positive rates of the detector on the original and noisy distributions, respectively, assuming the fake distribution as the positive class. Then, by definition,

$$\mathsf{AUROC}_{\mathcal{N}}(D) = \int_0^1 \mathsf{TPR}_{\mathcal{N}} \, d\mathsf{FPR}_{\mathcal{N}}.$$

Now, to relate this to AUROC, we lower bound $\mathsf{TPR}_{\mathcal{N}}$ and upper bound $\mathsf{FPR}_{\mathcal{N}}$ in terms of TPR and FPR.

$$
\begin{aligned}
\mathsf{TPR}_{\mathcal{N}} &= \mathbb{P}_{x \sim \mathcal{F}, \tilde{\phi} \sim \mathcal{N}(\phi(x), \sigma)}[D(\tilde{\phi}) = 1] \\
&= \mathbb{P}_{x \sim \mathcal{F}, \tilde{\phi} \sim \mathcal{N}(\phi(x), \sigma)}[D(\tilde{\phi}) = 1 | D(\phi(x)) = 1] \mathbb{P}_{x \sim \mathcal{F}}[D(\phi(x)) = 1] \\
&\quad + \mathbb{P}_{x \sim \mathcal{F}, \tilde{\phi} \sim \mathcal{N}(\phi(x), \sigma)}[D(\tilde{\phi}) = 1 | D(\phi(x)) = 0] \mathbb{P}_{x \sim \mathcal{F}}[D(\phi(x)) = 0] \\
&\qquad\qquad\qquad\qquad\qquad\qquad\qquad\qquad\qquad\qquad \text{(law of total probability)} \\
&\geq (1 - \alpha) \mathbb{P}_{x \sim \mathcal{F}}[D(\phi(x)) = 1] \qquad\qquad\qquad\qquad \text{(from Equation 3)} \\
&= (1 - \alpha) \mathsf{TPR}.
\end{aligned}
$$

$$
\begin{aligned}
\mathsf{FPR}_{\mathcal{N}} &= \mathbb{P}_{x \sim \mathcal{R}, \tilde{\phi} \sim \mathcal{N}(\phi(x), \sigma)}[D(\tilde{\phi}) = 1] \\
&= \mathbb{P}_{x \sim \mathcal{R}, \tilde{\phi} \sim \mathcal{N}(\phi(x), \sigma)}[D(\tilde{\phi}) = 1 | D(\phi(x)) = 1] \mathbb{P}_{x \sim \mathcal{R}}[D(\phi(x)) = 1] \\
&\quad + \mathbb{P}_{x \sim \mathcal{R}, \tilde{\phi} \sim \mathcal{N}(\phi(x), \sigma)}[D(\tilde{\phi}) = 1 | D(\phi(x)) = 0] \mathbb{P}_{x \sim \mathcal{R}}[D(\phi(x)) = 0] \\
&\qquad\qquad\qquad\qquad\qquad\qquad\qquad\qquad\qquad\qquad \text{(law of total probability)} \\
&\leq \mathbb{P}_{x \sim \mathcal{R}}[D(\phi(x)) = 1] \\
&\quad + (1 - \mathbb{P}_{x \sim \mathcal{R}, \tilde{\phi} \sim \mathcal{N}(\phi(x), \sigma)}[D(\tilde{\phi}) = 0 | D(\phi(x)) = 0]) \mathbb{P}_{x \sim \mathcal{R}}[D(\phi(x)) = 0] \\
&\leq \mathsf{FPR} + \alpha \mathbb{P}_{x \sim \mathcal{R}}[D(\phi(x)) = 0] \qquad\qquad\qquad \text{(from Equation 3)} \\
&\leq \mathsf{FPR} + \alpha.
\end{aligned}
$$

Therefore,

$$
\begin{aligned}
\mathsf{AUROC}_{\mathcal{N}}(D) &= \int_0^1 \mathsf{TPR}_{\mathcal{N}} \, d\mathsf{FPR}_{\mathcal{N}} \\
&\geq \int_0^1 (1 - \alpha) \mathsf{TPR} \, d\mathsf{FPR}_{\mathcal{N}} \qquad\qquad (\mathsf{TPR}_{\mathcal{N}} \geq (1 - \alpha)\mathsf{TPR}) \\
&= (1 - \alpha) \int_0^1 \mathsf{TPR} \, d\mathsf{FPR}_{\mathcal{N}} \\
&\geq (1 - \alpha) \int_0^{1 - \alpha} \mathsf{TPR} \, d\mathsf{FPR} \qquad\qquad (\mathsf{FPR}_{\mathcal{N}} \leq \mathsf{FPR} + \alpha) \\
&\geq (1 - \alpha) \left( \int_0^1 \mathsf{TPR} \, d\mathsf{FPR} - \alpha \right) \\
&= (1 - \alpha)(\mathsf{AUROC} - \alpha).
\end{aligned}
$$

Hence,

$$\mathsf{AUROC}(D) \leq \frac{\mathsf{AUROC}_{\mathcal{N}}(D)}{1 - \alpha} + \alpha.$$

$\square$

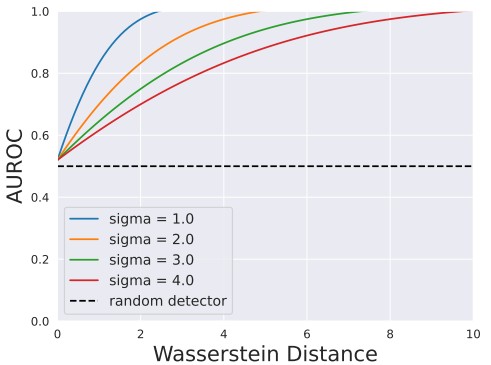
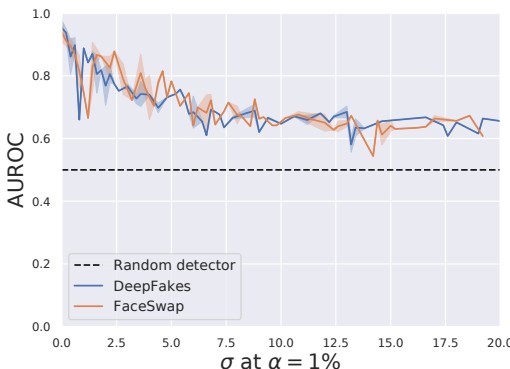

Figure 16: Deepfake detection performance bound w.r.t Wasserstein distance between real $\mathcal{R}$ and fake $\mathcal{F}$ distributions for different values of $\sigma$. A more robust detector (higher $\sigma$) has a lower performance.

Figure 17: AUROC vs. $\sigma$ plot for a $(\sigma, \alpha = 0.01)$-robust deep fake detector with a VGG-16-BN backbone on DeepFakes (deepfakes) and FaceSwap (MarekKowalski) datasets. Consistent with Theorem 2, AUROC drops as the robustness of the detector increases.

# E    PROOF OF LEMMA 3

**Statement.** *The total variation between the noisy distributions $\mathcal{R}_{\mathcal{N}}^{\phi}$ and $\mathcal{F}_{\mathcal{N}}^{\phi}$ is bounded by the Wasserstein distance of the original distributions $\mathcal{R}$ and $\mathcal{F}$ as follows:*

$$\mathsf{TV}(\mathcal{R}_{\mathcal{N}}^{\phi}, \mathcal{F}_{\mathcal{N}}^{\phi}) \leq \psi_{\sigma}\big(\mathsf{W}_{\phi}(\mathcal{R}, \mathcal{F})\big).$$

*Proof.* By definition of total variation, we have:

$$
\begin{aligned}
\mathsf{TV}(\mathcal{R}_{\mathcal{N}}^{\phi}, \mathcal{F}_{\mathcal{N}}^{\phi}) &= \sup_{E} \left| \mathbb{P}_{\tilde{\phi}_1 \sim \mathcal{R}_{\mathcal{N}}^{\phi}}[\tilde{\phi}_1 \in E] - \mathbb{P}_{\tilde{\phi}_2 \sim \mathcal{F}_{\mathcal{N}}^{\phi}}[\tilde{\phi}_2 \in E] \right| \\
&= \sup_{E} \Big| \mathbb{P}_{x_1 \sim \mathcal{R}, \tilde{\phi}_1 \sim \mathcal{N}(\phi(x_1), \sigma)}[\tilde{\phi}_1 \in E] \\
&\qquad - \mathbb{P}_{x_2 \sim \mathcal{F}, \tilde{\phi}_2 \sim \mathcal{N}(\phi(x_2), \sigma)}[\tilde{\phi}_2 \in E] \Big| \qquad \text{(definition of } \mathcal{R}_{\mathcal{N}}^{\phi} \text{ and } \mathcal{F}_{\mathcal{N}}^{\phi}) \\
&= \sup_{E} \Big| \mathbb{P}_{(x_1, x_2) \sim \gamma^{*}, \tilde{\phi}_1 \sim \mathcal{N}(\phi(x_1), \sigma)}[\tilde{\phi}_1 \in E] \\
&\qquad - \mathbb{P}_{(x_1, x_2) \sim \gamma^{*}, \tilde{\phi}_2 \sim \mathcal{N}(\phi(x_2), \sigma)}[\tilde{\phi}_2 \in E] \Big| \quad \text{(since } \gamma^{*} \text{ has marginals } \mathcal{R} \text{ and } \mathcal{F}) \\
&= \sup_{E} \left| \mathbb{E}_{(x_1, x_2) \sim \gamma^{*}} \left[ \mathbb{P}_{\tilde{\phi}_1 \sim \mathcal{N}(\phi(x_1), \sigma)}[\tilde{\phi}_1 \in E] - \mathbb{P}_{\tilde{\phi}_2 \sim \mathcal{N}(\phi(x_2), \sigma)}[\tilde{\phi}_2 \in E] \right] \right| \\
&\leq \sup_{E} \mathbb{E}_{(x_1, x_2) \sim \gamma^{*}} \left| \mathbb{P}_{\tilde{\phi}_1 \sim \mathcal{N}(\phi(x_1), \sigma)}[\tilde{\phi}_1 \in E] - \mathbb{P}_{\tilde{\phi}_2 \sim \mathcal{N}(\phi(x_2), \sigma)}[\tilde{\phi}_2 \in E] \right| \\
&\qquad\qquad\qquad\qquad\qquad\qquad\qquad\qquad\qquad\qquad\qquad\qquad\qquad (\text{since } |a + b| \leq |a| + |b|) \\
&\leq \mathbb{E}_{(x_1, x_2) \sim \gamma^{*}} \left[ \mathsf{TV}\big(\mathcal{N}(\phi(x_1), \sigma), \mathcal{N}(\phi(x_2), \sigma)\big) \right] \text{ (by definition of total variation)} \\
&\leq \mathbb{E}_{(x_1, x_2) \sim \gamma^{*}} \left[ \psi_{\sigma}\big(\|\phi(x_1) - \phi(x_2)\|\big) \right] \qquad\qquad\qquad \text{(from Equation 11)} \\
&\leq \psi_{\sigma}\big(\mathbb{E}_{(x_1, x_2) \sim \gamma^{*}} \left[ \|\phi(x_1) - \phi(x_2)\| \right] \big) \\
&\qquad\qquad\qquad\qquad\qquad\qquad \text{(since } \psi_{\sigma} \text{ is concave and Jensen's inequality)} \\
&= \psi_{\sigma}\big(\mathsf{W}_{\phi}(\mathcal{R}, \mathcal{F})\big). \qquad\qquad\qquad \text{(from definition of } \gamma^{*} \text{ and Equation 10)}
\end{aligned}
$$

$\square$

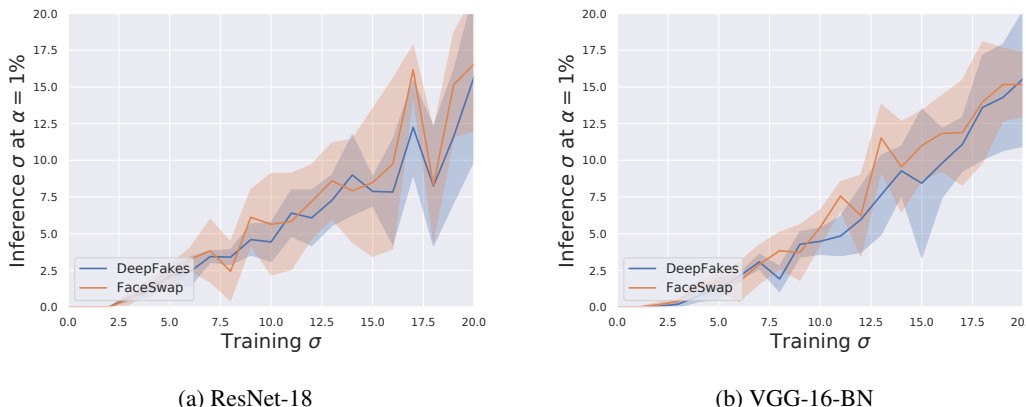

(a) ResNet-18                                    (b) VGG-16-BN

Figure 18: Detector robustness (inference $\sigma$ at $\alpha = 1\%$) to random noise in the $\phi$ latent space increases as the standard deviation of noise used for training increases. Various robust detectors are trained by adding Gaussian noise of standard deviation between 0 and 20 to the $\phi$ latent space. Y-axes represent the standard deviation of the noise at inference time on the test dataset for which the detector achieves $\alpha = 0.01$ as per Equation 3.

## F    MORE DETAILS ON DEEPFAKE DETECTOR EXPERIMENTS

Theorem 2 provides a robustness-reliability trade-off for deepfake detectors. Figure 8 shows how the AUROC reduces with robustness for different Wasserstein distances based on our bound. Figure 16 shows how the AUROC reduces with Wasserstein distance for various noise values $\sigma$. We perform experiments on the FaceForensics++ dataset hosted by Rössler et al. (2019) to empirically verify our theoretical insights. FaceForensics++ (Rössler et al., 2019) is a forensic dataset that consists of 1000 video sequences that are manipulated using different automated face manipulation techniques[1].For our experiments, we use frames from videos that are manipulated using FaceSwap (MarekKowalski) and Deepfakes (deepfakes). FaceSwap manipulations are based on classical computer graphics-based methods, while DeepFakes relies on a learning-based approach. We perform a set of preprocessing steps to extract aligned $228 \times 228$ face images from the videos using the Deep-Fakes software[2]. We randomly sample 5 frames from each video. We ensure that our final image datasets have no overlap of identities between the training and test splits. After preprocessing, our FaceSwap image dataset contains 4316 (1059, respectively) original and 3529 (1857, respectively) manipulated images in the training (test, respectively) dataset. Similarly, our DeepFakes image dataset contains 4316 (1059, respectively) original and 3522 (1843, respectively) manipulated images in the training (test, respectively) dataset.

We train different detectors with the standard deviation of noise $\sigma$ varied from 0 to 20 with the following objective

$$\min_{\theta} \ \frac{1}{N} \sum_{i=1}^{N} \ell(D(\phi(x_i) + n_i), y_i)$$

where $\ell$ is the cross-entropy loss, $n_i \sim \mathcal{N}(0, \sigma^2 I)$, and $\theta$ represent the parameters that defines $D$. For different detectors, we compute the inference $\sigma$ on the test dataset at which they achieve an $\alpha$ of 0.01 using Equation 3. Figure 18 shows that the detector robustness (inference $\sigma$ at $\alpha = 1\%$) to random noise increases as the training sigma increases. We use ten randomly sampled Gaussian noises for each sample $\phi(x)$ for this evaluation. Figures 9 and 17 plots the empirical trade-off between AUROC and robustness ($\sigma$ at $\alpha = 1\%$) for detectors with ResNet-18 and VGG-16-BN backbones, respectively, on the DeepFakes and FaceSwap datasets.

---

[1]https://github.com/ondyari/FaceForensics/
[2]https://github.com/deepfakes/faceswap

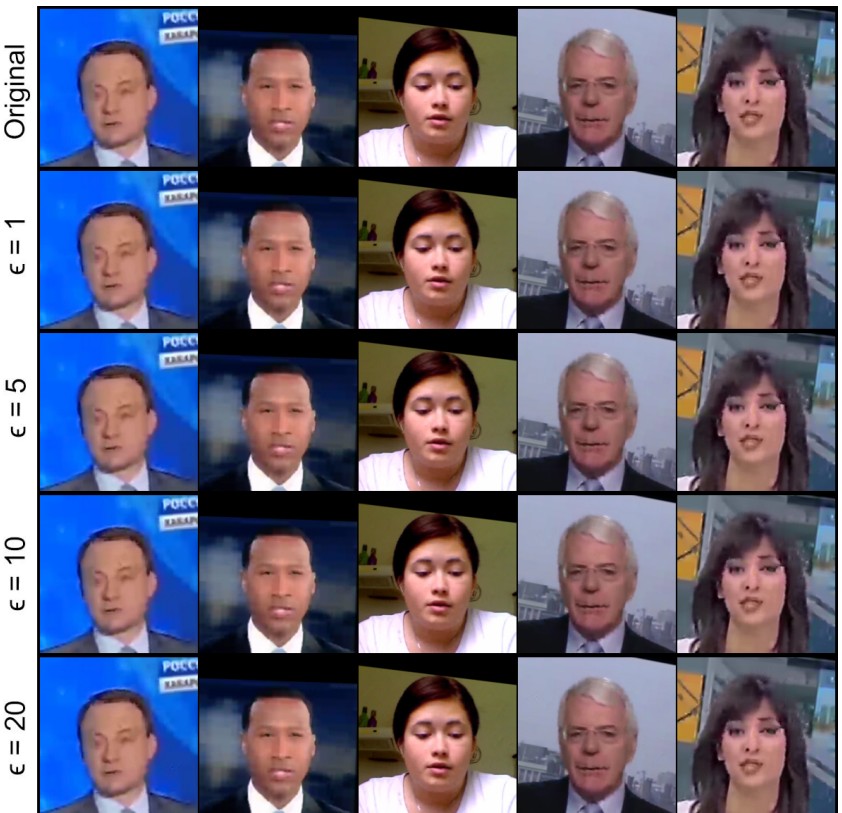

Figure 19: We use ResNet-18 and the FaceSwap dataset to visualize images that correspond to noisy latent space features. We optimize Equation 12 to find additive noises in the image space that cause large $\ell_2$ perturbations in the latent space $\phi$. In top row, we show the original images from the FaceSwap dataset. The rest of the rows show noisy images that produce perturbations corresponding $\epsilon$ in the latent space. Here, we show that small additive noises in the image space can lead to large perturbations in the $\phi$ space.

We also visualize how the noisy latent space vectors would look in the image space (see Figure 19). We optimize the following objective to find such images:

$$\min_{\delta} \ \left( \epsilon - \|\phi(x) - \phi(x + \delta)\|_2 \right)^2 \tag{12}$$

In the above optimization problem, we find an additive noise $\delta$ when added to a clean image $x$ leads to an $\ell_2$ perturbation of $\epsilon$ in the latent space. As shown in Figure 19, FaceSwap images with small perturbations in the input space can cause large perturbations in the latent space $\phi$ of ResNet-18.

