# OpenReview forum: "Robustness of AI-Image Detectors: Fundamental Limits and Practical Attacks"
_ICLR.cc/2024/Conference — ICLR 2024 poster_

### Official Review · Reviewer_APfc · 2023-10-28

**Soundness:** 2 fair
**Presentation:** 2 fair
**Contribution:** 1 poor
**Rating:** 5
**Confidence:** 4

**Summary:**

The paper analyzes the robustness of two common AI-generated image detection approaches: watermarking and classification. For watermarking, it points out that diffusion purification can effectively remove low-perturbation budget watermarks but fails to work on high-perturbation budget ones. For the latter one, the paper proposes using a model-substitution adversarial attack to remove the watermarks. It also proposes a spoofing attack against watermarking by blending a watermarked noise image with the non-watermarked target image. Finally, it demonstrates a trade-off between the robustness and reliability of classification-based deepfake detectors.

**Strengths:**

The paper lacks a clearly identifiable strength.

**Weaknesses:**

The paper is poorly written and not well-organized. For example, acronyms were defined far after their first usage. Figures are hard to follow and understand.

The contribution of the paper is unclear. It is common sense pointed out by several previous research that the stronger the perturbations, the more difficult for purification. Using random noise or adversarial perturbations to compromise the machine-learning-based forensics models (watermarking and deepfake detection) has been studied in the past, which is also mentioned in the related work section.

The experiment settings are insufficient to demonstrate the claimed problems. Besides DiffPure (Nie et al., 2022), there are other diffusion-based approaches, such as DDNM (Wang et al., 2022), and non-diffusion-based ones, which are not investigated.

The robustness of the proposed attack methods, which is an important property, was not evaluated. DiffPure (Nie et al., 2022), JPEG compression, or Gaussian blur can be used to mitigate such attacks, although it may slightly degrade the clean accuracy.

The paper should include a paragraph of ethics statement.

**Questions:**

Please refer to the comments in the weaknesses section.

---

> ### Author Response · Authors · 2023-11-21
>
> Thanks for your insightful comments. We hope the following responses can help address your concerns.
>
> > **Concern 1:** The paper is poorly written and not well-organized.
>
> We have updated several parts in the new version of the draft, in order to increase clarification and organization.
> - We added reference to the definition of terms that are formally defined in the paper after their usage, including $e_0, e_1$ in the introduction, and $R, F, R^t, F^t$ in Figure 1.
> - The Y-axis label of Figure 2 had a typo which is fixed now. New labels were added to Figure 1 to enhance readability.
> - New references were added to Section 3.1, to help the readers get familiar with the diffusion model terms that are being used in the section.
> - More information about the importance of Figure 2 was added to Section 3.1.
> - More explanation on the reason for the usage of denoising diffusion models in our theory was added to Section 3.1 and Appendix B. It is explained that other denoising techniques could also be used instead of denoising diffusion models, and references to related work were added.
> - A more detailed description of the spoofing attack, together with a pseudocode has been added to Appendix A.4.
> - In Section 4, formal definitions for some of the terms that are used in Theorem 2 were added to the main text.
>
> > **Concern 2:** The contribution of the paper is unclear.
>
> Our paper includes several noteworthy contributions.
>
> - Our first and perhaps most important contribution is the theoretical impossibility of designing reliable and imperceptible watermarks  (i.e., Theorem 1 in the paper). Lots of previous studies use image watermarks for different applications, such as detecting AI-made images, protecting copyrighted images, and establishing image ownership. In all but a few cases, these watermarking techniques utilize imperceptible perturbations. Our theoretical findings show the fundamental unreliability of such techniques and suggest a paradigm shift toward watermarks with higher perturbation budgets within the research community.
> - Our second contribution is the development of a novel model substitution black-box adversarial attack that breaks existing high perturbation budget watermarking techniques. This approach allows for attacking watermarking methods without requiring online query access to the models. To the best of our knowledge, the watermarking technique TreeRing has previously withstood black-box attacks in the literature, establishing itself as a reliable method. However, our attack successfully breaks this technique with an $\ell_\infty$ perturbation as small as 2/255.
> - Our third contribution is developing a spoofing attack against watermarking by adding a watermarked noise image to non-watermarked images to deceive the detector into flagging them as watermarked.
> - Our fourth and final contribution is to characterize a fundamental trade-off between the robustness and reliability of deepfake detectors and substantiate this concept through experiments.
>
>
> > **Concern 3:** Besides DiffPure (Nie et al., 2022), there are other diffusion-based approaches, such as DDNM (Wang et al., 2022), and non-diffusion-based ones, which are not investigated.
>
> We appreciate the reviewer's observation regarding the potential use of alternative denoising techniques to eliminate the introduced Gaussian noise from images. We have added this explanation to the new draft of the paper (in Section 3.1 and Appendix B), while citing some related work.
>
> Our theoretical findings establish an upper bound on the total variation of the distributions of watermarked and non-watermarked images by adding Gaussian noise to images. It is important to note that we do not rely on the denoising technique to calculate these bounds; rather, we employ them solely to generate high-quality output images. Consequently, any denoising technique, aside from DiffPure, can be employed to achieve the same theoretical bounds.
>
> It is true that a stronger denoising technique permits the use of a higher magnitude of Gaussian noise, resulting in a more substantial lower bound on the error according to our theory. Nonetheless, it's worth emphasizing that DiffPure has already proven its capability to empirically break existing imperceptible watermarks, showing the effectiveness of our proposed methodology.

---

> > ### Author Response · Authors · 2023-11-21
> >
> > > **Concern 4:** DiffPure (Nie et al., 2022), JPEG compression, or Gaussian blur can be used to mitigate such attacks
> >
> >
> > We appreciate your suggestion. Based on your comment, we conducted additional experiments. In the following tables, we present the AUROC of various watermarking techniques under both the DiffPure attack and our proposed adversarial attack. Post-attack mitigations, including DiffPure (t=0.2), JPEG compression, and Gaussian blur (k=5), are applied, with the understanding that they may marginally compromise clean accuracy.
> >
> > [AUROC under DiffPure attack with t=0.2]
> > | Method      | Base | Blur | JPEG | DiffPure |
> > | ----------- | ---- | ---- | ---- | -------- |
> > | WatermarkDM | **0.64** | 0.63 | 0.60 | 0.52 |
> > | MBRS        | 0.61 | **0.63** | 0.63 | 0.54 |
> > | RivaGAN     | 0.65 | **0.69** | 0.68 | 0.64 |
> > | DwtDct      | **0.54** | 0.54 | 0.54 | 0.54 |
> > | DwtDctSVD   | **0.56** | 0.56 | 0.56 | 0.56 |
> > | TreeRing    | **0.97** | 0.96 | 0.92 | 0.91 |
> > | StegaStamp  | 0.96 | 0.96 | **0.97** | 0.83 |
> >
> > [AUROC under DiffPure attack with t=0.3]
> > | Method      | Base | Blur | JPEG | DiffPure |
> > | ----------- | ---- | ---- | ---- | -------- |
> > | WatermarkDM | 0.57 | **0.60** | 0.56 | 0.56 |
> > | MBRS        | **0.54** | 0.49 | 0.42 | 0.54 |
> > | RivaGAN     | **0.58** | 0.57 | 0.56 | 0.56 |
> > | DwtDct      | **0.54** | 0.54 | 0.54 | 0.54 |
> > | DwtDctSVD   | **0.55** | 0.55 | 0.55 | 0.55 |
> > | TreeRing    | **0.93** | 0.91 | 0.87 | 0.88 |
> > | StegaStamp  | 0.78 | **0.80** | 0.77 | 0.66 |
> >
> > In the above table, the application of post-attack mitigations is not causing significant increases in the AUROC. This is anticipated since the primary aim of the DiffPure attack is the removal of watermarks from the watermarked images. Therefore, it is reasonable to expect that basic post-attack mitigations will encounter challenges in recovering the watermark.
> >
> >
> > [AUROC under adversarial attack with eps=4/255]
> > | Method      | Base | Blur (k=5) | Blur (k=15) | JPEG | DiffPure |
> > | ----------- | ---- | ---- | ---- | ---- | -------- |
> > | TreeRing | 0.03 | 0.02 | 0.02 | 0.05 | **0.89** |
> > | StegaStamp       | **1.0** | 1.0 | 1.0 | 0.99 | 0.87 |
> >
> > [AUROC under adversarial attack with eps=8/255]
> > | Method      | Base | Blur (k=5) | Blur (k=15) | JPEG | DiffPure |
> > | ----------- | ---- | ---- | ---- | ---- | -------- |
> > | TreeRing | 0.01 | 0.01 | 0.01 | 0.01 | **0.53** |
> > | StegaStamp       | **0.92** | 0.84 | 0.79 | 0.86 | 0.70 |
> >
> > [AUROC under adversarial attack with eps=12/255]
> > | Method      | Base | Blur (k=5) | Blur (k=15) | JPEG | DiffPure |
> > | ----------- | ---- | ---- | ---- | ---- | -------- |
> > | TreeRing | 0.01 | 0.01 | 0.01 | 0.01 | **0.07** |
> > | StegaStamp       | 0.49  | 0.42 | 0.34 | 0.49 | **0.56** |
> >
> > These results have been added to the new version of the paper, in Appendix A.5 (Tables 3 and 4).
> >
> > Please note that subsequent to the paper submission, some adjustments were made to the adversarial attack and substitute classifier training procedures, resulting in improved base results as shown in the tables. These changes include the configuration of warm-up attacks, and augmentations that are used for training the classifier.
> >
> > > **Concern 5:** The paper should include a paragraph of ethics statement.
> >
> > Thank you for your suggestion. We have added a paragraph at the end of the main text.

---

> > > ### Comment · Reviewer_APfc · 2023-11-22
> > >
> > > I would like to thank the authors for their responsiveness in addressing my inquiries and for revising the paper.
> > >
> > > > Our first and perhaps most important contribution is the theoretical impossibility of designing reliable and imperceptible watermarks (i.e., Theorem 1 in the paper). Lots of previous studies use image watermarks for different applications, such as detecting AI-made images, protecting copyrighted images, and establishing image ownership. In all but a few cases, these watermarking techniques utilize imperceptible perturbations. Our theoretical findings show the fundamental unreliability of such techniques and suggest a paradigm shift toward watermarks with higher perturbation budgets within the research community.
> > >
> > > The above statement, which is the most important contribution of the paper, is common knowledge in the watermarking community [A]. It is also common knowledge in the adversarial machine learning community that there is a trade-off between the attack ability and the imperceptibility of the perturbations. Recent work also demonstrated that invisible image watermarks are removable [B].
> > >
> > >
> > > > Our second contribution is the development of a novel model substitution black-box adversarial attack that breaks existing high perturbation budget watermarking techniques.
> > >
> > > The idea of using model substitution for black-box adversarial attacks is also not new [C].
> > >
> > >
> > > While I appreciate the effort put into the manuscript, because of the above reasons, regrettably, I must maintain my score. I wish the authors every success in their subsequent submission with their further improvements.
> > >
> > > *References:*
> > >
> > > [A] Agarwal, Namita, Amit Kumar Singh, and Pradeep Kumar Singh. "Survey of robust and imperceptible watermarking." Multimedia Tools and Applications 78 (2019): 8603-8633.
> > >
> > > [B] Zhao, Xuandong, Kexun Zhang, Zihao Su, Saastha Vasan, Ilya Grishchenko, Christopher Kruegel, Giovanni Vigna, Yu-Xiang Wang, and Lei Li. “Invisible Image Watermarks Are Provably Removable Using Generative AI.” arXiv, August 6, 2023.
> > >
> > > [C] Papernot, Nicolas, Patrick McDaniel, Ian Goodfellow, Somesh Jha, Z. Berkay Celik, and Ananthram Swami. "Practical black-box attacks against machine learning." Asia conference on computer and communications security, pp. 506-519. 2017.

---

> > > > ### Author Response · Authors · 2023-11-22
> > > >
> > > > We thank the reviewer for their quick reply.
> > > >
> > > > > The above statement, which is the most important contribution of the paper, is common knowledge in the watermarking community [A].
> > > >
> > > > It is straightforward that any type of machine learning model can be fooled by adding large enough perturbation to the data. However, for some tasks such as attacking watermarking techniques, the output images of the attack must have reasonable quality, and maintain similarity to the input image of the attack, which is something that diffusion purification is capable of. To the best of our knowledge, simple attacks, including the attacks discussed in [1], are not able to break recent state-of-the-art watermarking techniques. Another evidence that the "impossibility of designing imperceptible watermarks" is not common knowledge, is the fact that dozens of papers that design new imperceptible watermarks have been published in recent years (which we have cited in Section 2), and they have claimed robustness to existing attacks against watermarks, which consist of mostly common augmentations such as brightness, rotation, contrast, etc. Due to this lack of strong attacks against watermarking techniques, we are proposing diffusion purification as an attack that certifiably breaks existing imperceptible watermarks, and we suggest future watermarking techniques to evaluate their methods against diffusion purification, and our other attacks.
> > > >
> > > > > Recent work also demonstrated that invisible image watermarks are removable [B].
> > > >
> > > > We have already cited this work in Section 2 (i.e., Prior Work), and explained that this is a concurrent work that provides similar theoretical results on the utilization of diffusion purification for removing watermarks, even though they use a different technique to achieve their theoretical results. At the time of writing our paper, we were not aware of the results from [B]. Meanwhile, it is worth mentioning that [B] is not able to attack high-perturbation watermarks such as StegaStamp and TreeRing, and advocates for TreeRing and other "semantic watermarks" as reliable and robust watermarks. Our adversarial attack shows that this claim is not true, and raises the challenges in designing high-perturbation watermarks. In addition, we are providing a specialized attack to target the spoofing error of watermark detectors, which is something that has not been considered a threat in previous work.
> > > >
> > > > > The idea of using model substitution for black-box adversarial attacks is also not new.
> > > >
> > > > We agree with the reviewer that model substitution attacks have been used for other applications and purposes. However, the application of these attacks in the watermarking domain has not been investigated before, and we believe it is indeed a practical attack that is needed in the current literature. The goal of our paper is to investigate the usage of watermarks for flagging AI-generated images, and if a watermark is used for this task, many such watermarked AI-generated images will be available for public use. This makes our assumption of having access to watermarked samples for training a substitute model realistic.
> > > > In contrast, conventional black-box adversarial attacks require a significant number of online query accesses to the watermark injector model, just to produce a single adversarial sample.
> > > >
> > > > We remain open to further discussion and sincerely hope that our clarifications may positively influence the reviewer's perspective on our work.
> > > >
> > > >
> > > > #### [A] Agarwal, Namita, Amit Kumar Singh, and Pradeep Kumar Singh. "Survey of robust and imperceptible watermarking." Multimedia Tools and Applications 78 (2019): 8603-8633.
> > > >
> > > > #### [B] Zhao, Xuandong, Kexun Zhang, Zihao Su, Saastha Vasan, Ilya Grishchenko, Christopher Kruegel, Giovanni Vigna, Yu-Xiang Wang, and Lei Li. “Invisible Image Watermarks Are Provably Removable Using Generative AI.” arXiv, August 6, 2023.

---

> > > > > ### Comment · Reviewer_APfc · 2023-11-23
> > > > >
> > > > > I extend my gratitude once more to the authors for their prompt responses.
> > > > >
> > > > > Upon thorough reflection, I made a slight adjustment to the score. Despite this, achieving a positive rating remains challenging due to the high standards set by ICLR. Nevertheless, I wish the author continued success.

---

### Official Review · Reviewer_1Mes · 2023-10-30

**Soundness:** 3 good
**Presentation:** 3 good
**Contribution:** 3 good
**Rating:** 6
**Confidence:** 4

**Summary:**

This paper analyzes the robustness of various AI image detectors, including watermarking and classifier-based deepfake detectors, wherein watermarking is considered a promising method for identifying AI-generated images.

The paper also evaluates the trade-off between evasion error rate and spoofing error rate in watermarking methods, introducing subtle image perturbations. Besides, it is demonstrated that a diffusion purification attack that amplifies the error rates of low perturbation budget watermarking methods, thereby revealing the fundamental limits of the robustness of image watermarking methods. For large-perturbation watermarking methods, the diffusion purification attack is ineffective. Therefore, the authors propose a model substitution adversarial attack to successfully remove watermarks.

Overall, this paper makes significant contributions to the robustness of AI image detection methods, supported by detailed theoretical proofs of the viewpoints presented. Some of the theoretical analyses and attack methods in the article are relatively complex and may require readers to have a certain level of background knowledge to fully understand.

**Strengths:**

1) Originality: The paper challenges the limitations of existing watermarking techniques by proposing new attack methods, driving progress in the field.

2) Quality: The quality of the paper is very high, with in-depth and rigorous theoretical analysis, reasonable experimental design, and results that fully validate the theoretical analysis.

3) Clarity: The structure of the paper is clear, the logic is tight, and the discussion is detailed and easy to understand. However, some of the complex theoretical analyses and attack methods may require readers to have a certain level of background knowledge.
Importance: By revealing the vulnerabilities of existing watermarking techniques, the paper lays the groundwork for further research and development in the field. Additionally, by proposing new attack methods, the paper challenges the limitations of existing watermarking technologies and promotes progress in the field.

**Weaknesses:**

1) Complexity of Theoretical Analysis:
While the paper provides an in-depth theoretical analysis, the complexity of these analyses might pose challenges for some readers, especially those who are not familiar with diffusion models and the theoretical underpinnings of adversarial attacks. Certain sections of the paper may appear somewhat opaque to these readers. It would be beneficial if the authors could simplify the explanations or provide additional resources to aid understanding.


2) Complexity of Theoretical Analysis:
While the paper provides an in-depth theoretical analysis, the complexity of these analyses might pose challenges for some readers, especially those who are not familiar with diffusion models and the theoretical underpinnings of adversarial attacks. Certain sections of the paper may appear somewhat opaque to these readers. It would be beneficial if the authors could simplify the explanations or provide additional resources to aid understanding.

**Questions:**

1) Problem Description: The experiments conducted in the article primarily utilize the ImageNet dataset, potentially limiting the generalizability of the results.
Recommendation: In future work, conduct experiments using multiple datasets from various fields and sources to validate the effectiveness and stability of the method.

2) Problem Description: The theoretical analysis presented in the article is quite complex, which may be challenging for all readers to comprehend.
Recommendation: Provide more intuitive explanations and examples, and simplify some of the theoretical derivations to make them more accessible and easier to understand.

---

> ### Author Response · Authors · 2023-11-21
>
> Thanks for your helpful comments. We hope the following responses can help address your concerns.
>
> >  **Weakness 1:** It would be beneficial if the authors could simplify the explanations or provide additional resources to aid understanding.
>
> In the new version of our draft, we have added a reference to [1] (in Section 3.1) which provides comprehensive definitions for diffusion models and the terms that are used in our paper. Most of the terms that we use in our work such as $x_0$, $x_t$, $\bar{\alpha}_t$ are consistent with this survey paper.
>
> >  **Question 2:** Provide more intuitive explanations and examples, and simplify some of the theoretical derivations to make them more accessible and easier to understand.
>
> To enhance the clarity of the theoretical parts of the paper, we have added more explanations to Sections 3 and 4, and the appendix. We added the formal definition of the terms used in Theorem 2 to the main text of Section 4. Additionally, we clarified the reason for the usage of denoising diffusion models in our theory, and how it can be replaced with other denoising techniques (Section 3.1, and Appendix B).
>
> > **Question 1:** In future work, conduct experiments using multiple datasets from various fields and sources to validate the effectiveness and stability of the method.
>
> We agree with the reviewer’s comment that including more datasets could support the generalizability of our work. Adding new datasets can be time-consuming, especially since some of the watermarking methods require training of their models on the target dataset. Nonetheless, it will be one of our priorities to include more datasets such as COCO and CIFAR in our work, hopefully for the final draft of the paper.
>
>
>
>  [1] Understanding Diffusion Models: A Unified Perspective, Calvin Luo 2022

---

### Official Review · Reviewer_h5mn · 2023-11-05

**Soundness:** 3 good
**Presentation:** 3 good
**Contribution:** 2 fair
**Rating:** 6
**Confidence:** 3

**Summary:**

This work investigates the resilience of AI-image detection methods, focusing on watermarking and classifier-based deepfake detectors. The authors highlight the crucial need to distinguish between authentic and AI-generated content due to the rising threat of fake materials being used as genuine ones. They reveal a fundamental trade-off in the effectiveness of watermarking techniques, showcasing the limitations of low-perturbation and high-perturbation watermarking methods. Specifically, they propose diffusion purification as a certified attack against low-perturbation watermarks and a model substitution adversarial attack against high-perturbation watermarks. Additionally, the paper emphasizes the vulnerability of watermarking methods to spoofing attacks, which can lead to the misidentification of authentic images as watermarked ones. Finally, the authors extend their analysis to classifier-based deepfake detectors, demonstrating a trade-off between reliability and robustness.

**Strengths:**

The strengths of this work are as follows:
1. Comprehensive analysis: The paper provides a comprehensive analysis of the robustness of AI-image detection methods, focusing on both watermarking and classifier-based deepfake detectors. This thorough investigation helps in understanding the limitations and vulnerabilities of these methods.
2. Practical attacks: The paper introduces practical attacks, such as diffusion purification and model substitution adversarial attacks, to illustrate the vulnerabilities of different watermarking methods.
3. Clarity in trade-offs: The paper effectively highlights the trade-offs between various aspects of AI-image detection methods, such as the trade-off between evasion error rate and spoofing error rate in the case of watermarking methods. This clarity helps in understanding the challenges associated with designing robust AI-image detection systems.
4. Sound theoretical study and guidelines for designing robust watermarks: The paper offers insights into the attributes that a robust watermark should possess, including significant perturbation, resistance to naive classification, and resilience to noise from other watermarked images. These guidelines can serve as a valuable reference for researchers and developers working on improving the security and reliability of AI-image detection methods.

**Weaknesses:**

The experimental results are missing a key element, specifically the PSNR, SSIM, or other image quality metrics comparing the diffusion-purified or adversarially attacked images with the original images. This result is crucial as adversaries aim to eliminate watermarks while preserving high-quality images simultaneously.

**Questions:**

Please refer to the weakness.

---

> ### Author Response · Authors · 2023-11-21
>
> Thank you for your positive feedback. We hope the following response can help address your concern.
>
> > **Weakness 1:** The experimental results are missing a key element, specifically the PSNR, SSIM, or other image quality metric
>
> We appreciate your valuable suggestion, which was an aspect we overlooked in our experiments. Below, you'll find the PSNR and SSIM values for images subjected to both DiffPure and adversarial attacks. In the case of the DiffPure attack, the output images with t=0.2 exhibit reasonable quality, while the attack significantly lowers the AUROC for imperceptible watermarking techniques, as shown in Figure 3 of the paper. As for the adversarial attack, the epsilon values employed in the paper align with standard perturbation budgets in the adversarial robustness literature. As indicated in the tables, these values result in images with reasonable quality. We have added these tables to the appendix of our paper.
>
> [mean PSNR of images attacked with DiffPure w.r.t t]
> | Method      | t=0.1 | t=0.2 | t=0.3 |
> | ----------- | ---- | ---- | ---- |
> | WatermarkDM | 30.33 | 26.41 | 23.87|
> | MBRS        | 29.96 | 26.23 | 23.76 |
> | RivaGAN     | 29.77 | 26.10| 23.61|
> | DwtDct      | 29.64| 26.03  | 23.70 |
> | DwtDctSVD   | 29.69 | 26.08| 23.60|
> | TreeRing    | 32.45 | 28.27 | 25.49 |
> | StegaStamp  | 30.35 | 26.52| 24.08|
>
> [mean SSIM of images attacked with DiffPure w.r.t t]
> | Method      | t=0.1 | t=0.2 | t=0.3 |
> | ----------- | ---- | ---- | ---- |
> | WatermarkDM | 0.86 | 0.75| 	0.66 |
> | MBRS        | 0.83 | 0.73| 0.64|
> | RivaGAN     | 0.83| 0.72 |0.63|
> | DwtDct      | 0.83| 0.72 | 0.63|
> | DwtDctSVD   | 0.83|0.72|0.63|
> | TreeRing    | 0.92 | 0.86| 0.81|
> | StegaStamp  | 0.84|0.73|0.64|
>
> [mean PSNR of adversarially attacked images w.r.t epsilon]
> | Method      | eps=4 | eps=8 | eps=12 |
> | ----------- | ---- | ---- | ---- |
> | TreeRing    | 36.60 | 32.41 | 30.18|
> | StegaStamp  | 36.11  |31.20|28.20|
>
> [mean SSIM of adversarially attacked images w.r.t epsilon]
> | Method      | eps=4 | eps=8 | eps=12 |
> | ----------- | ---- | ---- | ---- |
> | TreeRing    | 0.96 | 0.90 |0.86|
> | StegaStamp  | 0.95 |0.86|0.75|

---

> > ### Comment · Reviewer_h5mn · 2023-11-22
> > **Response to Authors**
> >
> > Thanks for your efforts in addressing my concerns. Based on the responses and evaluation of the work, I would like to maintain my rating.

---

### Official Review · Reviewer_pjit · 2023-11-07

**Soundness:** 3 good
**Presentation:** 4 excellent
**Contribution:** 3 good
**Rating:** 6
**Confidence:** 4

**Summary:**

In this paper, the authors introduce a diffusion purification attack to break AI-image detection methods using watermarking with a low perturbation budget. For high perturbation image watermarking, they develop a model substitution adversarial attack. Besides, they successfully implemented a spoofing attack by adding a watermarked noise image with non-watermarked ones. Furthermore, they use comprehensive experiments to substantiate the trade-off between robustness and reliability of deepfake detectors.

**Strengths:**

A novel watermarking erasing method is proposed to break high perturbation budget watermarking like Tree Ring or StegaStamp.
This paper is well-written and easy to understand. The presentation and organization of the paper are good.

**Weaknesses:**

1.	The crux of this paper is theorem 1, which gives a lower bound for the sum of evasion and spoofing errors with regard to the Wasserstein distance between diffusion purification processed images. However, the empirical studies are not sufficient.
2.	The empirical studies for theorem 2 are unpractical. It is almost impossible to add noise to the interior feature maps directly in practical cases.

**Questions:**

1. In Fig. 6, authors adopt four low perturbation budget watermarking, including DWTDCT, DWTDCTSVD, RivaGAN, WatermarkDM, to validate theorem 1. More watermarking methods are required to be considered, like reference [1]. Many cutting-edge watermarking methods are absent in Fig. 6.
2. The experimental setup of section 4 is impractical. I suggest authors add noise to spatial images directly instead of interior feature maps.
3. The title of the paper is “AI-image detectors”. The authors only consider fake facial images in section 4. AI-generated facial detectors are only a small part of AI-image detectors. Therefore, more AI-image detectors for arbitrary contexts require consideration.

[1] MBRS : Enhancing Robustness of DNN-based Watermarking by Mini-Batch of Real and Simulated JPEG Compression

---

> ### Author Response · Authors · 2023-11-21
>
> Thanks for your valuable comments. We hope the following responses can help address your concerns.
>
> > **Question 1:** More watermarking methods are required to be considered.
>
> We are thankful for the reviewer's suggestion. We have included results for the MBRS method in our paper, as recommended, and we are working on adding more methods such as StableSignature [1] and HiDDeN [2] to the final draft. We would like to highlight that our DiffPure attack is not just an empirical attack; it comes with a theoretical guarantee (Theorem 1) establishing a clear trade-off between type I and type II errors for any imperceptible watermarking methods. This, in our view, reduces the urgency for an exhaustive evaluation of every imperceptible watermarking technique that currently exists.
>
> > **Question 2:**  The experimental setup of section 4 is impractical. I suggest authors add noise to spatial images directly instead of interior feature maps.
>
> Thank you for the comment. In Appendix F (Figure 19), we add new experiments (in purple text) where noise is added to the images. This result shows that adding small noises to the image space can cause high $\ell_2$ perturbations in the latent space. Also, note that the tradeoff we present in Figure 8 (Theorem 2) will be more significant as AI images become more realistic and the Wasserstein distance between real and fake distributions decreases. This means that it will become easier to affect the performance of image detectors by adding lower levels of noise to the images as generative image models evolve.
>
> > **Question 3:** More AI-image detectors for arbitrary contexts require consideration.
>
> Our theory in Section 4 is general and holds for any real and fake distributions. We consider an application of deepfakes and perform our experiments on facial images since they are popular in the research community. We take your feedback positively and are working on adding more experiments on a different dataset. We will update the manuscript if we finish our experiments before the discussion period ends. Otherwise, we will add these results to the final version of the paper.
>
>
> [1] The Stable Signature: Rooting Watermarks in Latent Diffusion Models, Fernandez et. al., 2023
>
> [2] HiDDeN: Hiding Data With Deep Networks, Zhu et. al., 2018

---

### Meta-Review · Area_Chair_3GT6 · 2023-12-06

**Metareview:**

In this paper, authors analyze the robustness of different AI-image detectors including watermarking and classifier-based  deepfake detectors. The main contributions are: 1) they found a fundamental trade-off between evasion and spoofing error rates of image watermarking upon the application of a diffusion purification attack; 2) they show that diffusion purification attack can break a whole range of watermarking methods that introduce subtle image perturbations; 3) For high perturbation image watermarking, they developed a model substitution adversarial attack that can successfully remove the watermarks; 4) they introduced a spoofing attack against watermarking by adding a watermarking noise image to clean images; 5) They developed a fundamental trade-off between the robustness and reliability of deepfake detectors. Strengths and weaknesses given by reviewers are: 1) paper is well written and easy to follow; 2) comprehensive analysis of robustness of AI-image detection methods;3) proposed method are novel. Weaknesses are: 1) more comparison needed; 2) some additional experimental results are needed.

The reviewers' score are: 3 "6: marginally above the acceptance threshold", and 1 5: marginally below the acceptance threshold. In the rebuttal authors added a lot of experiments to address reviewers' concerns.

**Justification For Why Not Higher Score:**

lacking additional experiments to convince reviewers to raise their rating.

**Justification For Why Not Lower Score:**

positive feedback from reviewers and this is important problems for the community to consider.

---

### Decision · Program_Chairs · 2024-01-16

Accept (poster)